# Stabilizing Heterogeneous Federated Learning via Feature Decorrelation and Bilateral Alignment

## Abstract

Data heterogeneity poses a major challenge in federated learning, leading to significant degradation in global model performance. Prior studies have shown that heterogeneity induces dimensional collapse and biased classifiers, which hinder the learning of both feature extractors and classifiers. To tackle these issues, existing approaches apply feature decorrelation to mitigate dimensional collapse and adopt a synthetic classifier with a projector to reduce classifier bias. However, these decorrelation methods fail to prevent small singular values from collapsing to zero, slowing the mitigation of dimensional collapse. Besides, the synergy among the feature extractor, projector and synthetic classifier is overlooked, leading to divergent optimization across clients. To overcome these limitations, we propose **FedBlade**, a **fed**erated learning framework with **bi**lateral **a**lignment and feature **de**correlation. Our feature decorrelation method accelerates the mitigation of dimensional collapse by yielding exponential gradients, while the bilateral alignment method enhances synergy among model modules and ensures consistency across clients. Extensive experimental results demonstrate that FedBlade outperforms relevant baselines and achieves faster convergence of the global model.

## 1 Introduction

Federated learning (FL) (McMahan et al., 2017) is a decentralized paradigm that trains a global model across multiple clients without sharing raw data. As privacy concerns grow, FL has attracted significant attention. A major challenge in FL is data heterogeneity, which arises from discrepancies in the local data distributions across clients. In particular, this work focuses on the label skew setting, where the label distribution differs across clients.

Recent work has explored various approaches to address label skew, including regularization (Li et al., 2020; Acar et al., 2021), optimization (Reddi et al., 2020), model aggregation (Hsu et al., 2019; Ye et al., 2023b), feature alignment (Li et al., 2021; Tan et al., 2022; Ye et al., 2023a) and classifier calibration (Luo et al., 2021; Zhou et al., 2023). Beyond these directions, Shi et al. (2023) reveal that both local and global models suffer from dimensional collapse under label skew, where representations concentrate in a subspace rather than spanning the full representation space. This collapse severely degrades model generalization. To address it, Shi et al. (2023) propose FedDecorr, a regularization term that encourages representations to occupy the full ambient space. Specifically, FedDecorr minimizes the Frobenius norm of the representation correlation matrix, thereby discouraging the tail singular values of the representation covariance matrix from collapsing to zero. However, the gradient of FedDecorr is linear, which limits its ability to penalize small singular values and hinders the recovery of the ambient representation space.

Another problem induced by heterogeneous data is classifier bias. Luo et al. (2021) find that classifier layers exhibit greater bias than representation layers, and Zhou et al. (2023) show that such bias creates a vicious cycle between misaligned features and biased classifiers across clients. FedUV (Son et al., 2024) applies two regularizers on pairwise features and logits, aiming to enlarge prediction variance and prevent classifier degeneration. Unlike methods that target dimensional collapse in the representation space, FedUV focuses on the singular values of the classifier weight matrix.

Recent works (Li et al., 2023; Xiao et al., 2024) have investigated mitigating classifier bias by introducing a fixed and synthetic equiangular tight frame (ETF) classifier shared across clients. The ETF classifier enforces feature prototypes to converge to an optimal structure with maximal pairwise angles (Papyan et al., 2020; Yang et al., 2022). To encourage features to collapse into the ETF structure, FedETF (Li et al., 2023) employs a projector that maps raw features into a space where neural collapse is more likely to emerge. Thus, FedETF consists of three key modules: a feature extractor, a projector, and an ETF classifier. However, FedETF overlooks the synergy among these modules, leading to mismatches between projected features and the ETF classifier.

These two issues arise from distinct modules, i.e., the feature extractor and projector. We highlight two key challenges concerning these two modules:

`C1`: *How can we amplify gradients with respect to small singular values of representation covariance matrix?*

FedDecorr promotes decorrelation by penalizing the Frobenius norm of the correlation matrix, but its uniform treatment of entries yields linear gradients that fail to strongly penalize small singular values, leaving dimensional collapse insufficiently mitigated. To address this issue, it is important to yield larger gradients for the small singular values. Motivated by this intuition, we propose LDDecorr, a spectrum-aware feature decorrelation method that maximizes the log-determinant of the correlation matrix. As analyzed in Sec. 4.1, LDDecorr yields exponential gradients that impose infinite penalty on small singular values, thereby preventing dimensional collapse more effectively than FedDecorr.

`C2`: *How can we ensure coherent alignment among the feature extractor, projector, and ETF classifier?*

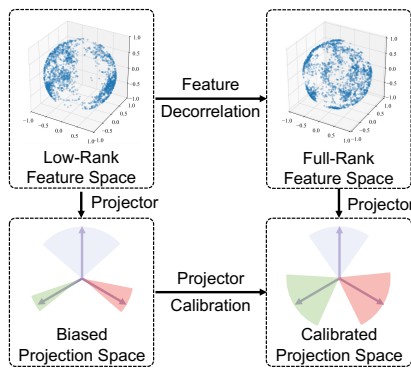

Figure 1: **Problem illustration.** Under label skew, FL faces two key issues: (1) dimensional collapse, where features concentrate in a low-rank subspace; and (2) projector bias toward head classes, which misaligns projected features with the ETF classifier.

Although the ETF classifier is fixed and shared across clients, the bias of the projector is overlooked. Inspired by feature alignment methods (Tan et al., 2022; Ye et al., 2023a), the global prototypes provide a uniform information, which can also be used to align the projector across clients. Besides, the prototypes serve as the bridges among the feature extractor, projector and ETF classifier, enhancing the synergy among these modules. Building on this idea, we propose PBA, a prototype-guided bilateral alignment method. PBA uses global prototypes to align the feature extractor and projector simultaneously during local training, ensuring that the feature spaces are consistent across clients and projected prototypes are close to corresponding ETF classifiers. As a result, the feature extractor, projector, and ETF classifier become colinear under PBA.

LDDecorr and PBA address *distinct yet interdependent challenges* under label skew. LDDecorr prevents dimensional collapse and preserves expressive capacity, which is crucial for generating informative prototypes in PBA. Conversely, PBA imposes structured ETF-like geometry that counteracts potential side effects of strong decorrelation, ensuring that the rank-expansion benefits of LDDecorr translate into improved class separation. Finally, we propose **FedBlade**, a **fed**erated learning framework with **b**ilateral **a**lignment and feature **de**correlation. With the help of these two components, the inter-class separation is increased and intra-class variance is reduced, enforcing the formation of neural collapse. Our main contributions are summarized as follows.

- We revisit the feature decorrelation term in federated learning, and propose LDDecorr, a spectrum-aware feature decorrelation method that enhances the mitigation of dimensional collapse. LDDecorr produces exponential gradients, imposing an infinite penalty on small singular values (Sec. 4.1).

- We propose PBA, a bilateral alignment method that simultaneously calibrates the feature extractor and projector. PBA enforces the synergy among the feature extractor, projector and ETF classifier (Sec. 4.2).

- We propose FedBlade, a federated learning framework with bilateral alignment and feature decorrelation. By pulling representations toward their corresponding ETF directions, PBA provides a structured geometric anchor that counteracts the potential side effects of strong feature decorrelation, while preserving the intended rank-expansion benefits of LDDecorr. Experimental results demonstrate that FedBlade outperforms relevant baselines.

## 2 RELATED WORK

### 2.1 LABEL SKEW IN FEDERATED LEARNING

Federated learning (McMahan et al., 2017) is a decentralized machine learning paradigm enabling training a global model without sharing raw training data. However, federated learning suffers from unstable convergence caused by data heterogeneity. One major challenge of data heterogeneity is label skew. To tackle this challenge, recent works have investigated a variety of solutions, such as regularization (Li et al., 2020; Karimireddy et al., 2020; Acar et al., 2021), optimization (Reddi et al., 2020), model aggregation (Hsu et al., 2019; Wang et al., 2020b; Ye et al., 2023b), batch normalization (Wang et al., 2023; Zhong et al., 2024; Zhang et al., 2024a), feature alignment (Li et al., 2021; Tan et al., 2022; Ye et al., 2023a; Guo et al., 2023; Zhang et al., 2024b), logits calibration (Zhang et al., 2022), and classifier calibration (Luo et al., 2021; Oh et al., 2022; Zhou et al., 2023; Guo et al., 2023; Son et al., 2024). In particular, this paper focuses on the classifier bias caused by label skew. Luo et al. (2021) find that classifier bias is greater than in other layers, and propose a federated learning method calibrating the classifier with virtual features after training. To further address classifier bias, Li et al. (2023) propose FedETF, which employs a synthetic simplex ETF as a fixed classifier shared across all clients. This design implicitly encourages clients to learn a unified representation space. However, the projector itself may still be biased, leading to unstable model convergence.

### 2.2 DIMENSIONAL COLLAPSE

Dimensional collapse is a phenomenon primarily studied in self-supervised learning (SSL) (Ermolov et al., 2021; Hua et al., 2021; Jing et al., 2022; He et al., 2024), where learned representations concentrate in a low-rank subspace and lose per-dimension variance. From a spectral perspective, dimensional collapse is characterized by a few dominant singular values while the remaining singular values shrink toward zero. Jing et al. (2022) formalize this problem in SSL and analyze how projection heads interact with the singular value spectrum of the embedding space. A line of works Zbontar et al. (2021); Bardes et al. (2021) address dimensional collapse by explicitly spreading variance and reducing redundancy across feature dimensions. He et al. (2024) introduce orthogonality regularization, mitigating dimensional collapse in representations, hidden features, and weight matrices. Beyond SSL, dimensional collapse has also been observed in federated learning, where stronger client heterogeneity exacerbates this problem. To counter this, FedDecorr (Shi et al., 2023) introduces decorrelation regularization, and Seo et al. (2024) propose a relaxed contrastive learning loss to avoid collapsed representations when incorporating supervised contrastive learning in federated learning. However, FedDecorr only yields linear gradients, which is insufficient to prevent the small singular values from collapsing to zero.

### 2.3 NEURAL COLLAPSE

Neural Collapse (NC) describes a terminal-phase geometry in supervised classification. Empirically, within-class features concentrate at their means, those means arrange as a simplex equiangular tight frame, and last-layer weights align with the means (Papyan et al., 2020). Subsequent analyses under squared loss make neural collapse amenable to theory via the central path description of gradient flow (Han et al., 2022) and global-optimality results in the unconstrained features model (Zhou et al., 2022). Tirer & Bruna (2022) extend the unconstrained features model with depth and regularization, and Súkeník et al. (2023) establish neural collapse in multi-layer settings. Although within-class concentration persists, ETF structure can deform and weight–mean alignment becomes sample-size dependent, motivating approaches that enforce ETF-like classifiers (Yang et al., 2022; Hong & Ling, 2024). Building on this idea, Li et al. (2023) mitigate classifier bias and feature misalignment in federated learning by introducing a fixed ETF classifier.

# 3 PRELIMINARIES

## 3.1 FEDERATED LEARNING

In this paper, we consider a federated learning setting with $K$ clients and a central server. Considering a classification task with $C$ classes, each client $k$ owns a local training dataset $D_k = \{\boldsymbol{x}_i, y_i\}_{i=1}^{n_k}$, where $n_k = \sum_{c=1}^{C} n_k^c$ denotes the number of samples. Under data heterogeneity setting, the data distribution $P(\mathcal{X}, \mathcal{Y})$ varies across clients, where $\mathcal{X}$ is the input space and $\mathcal{Y}$ is the label space. In particular, this paper focuses on label skew, where the label marginal distribution $P(\mathcal{Y})$ varies across clients, i.e., $P_i(\mathcal{Y}) \neq P_j(\mathcal{Y})$ for two clients $i$ and $j$. The goal of federated learning is to collaboratively train a global model without sharing raw training data. The local objective is $F_k := \mathbb{E}_{(\boldsymbol{x},y) \sim D_k}[\mathcal{L}(\boldsymbol{w}; \boldsymbol{x}, y)]$ and the global objective can be formulated as:

$$\min_{\boldsymbol{w} \in \mathbb{R}^d} \left\{ F(\boldsymbol{w}) := \sum_{k=1}^{K} \frac{n_k}{n} F_k(\boldsymbol{w}) \right\}, \tag{1}$$

where $n = \sum_{k=1}^{K} n_k$ and $\mathcal{L}$ is the loss function. We decompose the model into a feature extractor $f_{\boldsymbol{\theta}}$ and a classifier $f_{\boldsymbol{\phi}}$, which are parameterized by $\boldsymbol{\theta}$ and $\boldsymbol{\phi}$, respectively. The feature extractor $f_{\boldsymbol{\theta}} : \mathcal{X} \to \mathcal{Z}$ maps the input $\boldsymbol{x}$ into a feature vector $\boldsymbol{z} = f_{\boldsymbol{\theta}}(\boldsymbol{x})$ in the feature space $\mathcal{Z} \in \mathbb{R}^d$. Then, the classifier $f_{\boldsymbol{\phi}}$ maps the feature vector $\boldsymbol{z}$ into the class space $\mathbb{R}^C$.

Our study follows the conventional federated learning mechanism FedAvg (McMahan et al., 2017). In round $t$, the server selects a group of clients $\mathcal{I}^{(t)}$ and sends the global model $\boldsymbol{w}$ to them. After local training, each selected client $k \in \mathcal{I}^{(t)}$ sends its local model $\boldsymbol{w}_k$ to the server, and the global model are aggregated as:

$$\boldsymbol{w}^{(t+1)} = \sum_{k \in \mathcal{I}^{(t)}} \frac{n_k}{\sum_{i \in \mathcal{I}^{(t)}} n_i} \boldsymbol{w}_k^{(t)}. \tag{2}$$

## 3.2 EQUIANGULAR TIGHT FRAME CLASSIFIER

Recent works (Li et al., 2023; Xiao et al., 2024) address classifier bias by employing a fixed and synthetic equiangular tight frame (ETF) classifier. The ETF design is inspired by neural collapse (NC) (Papyan et al., 2020), a phenomenon in which deep classifiers exhibit a set of geometric regularities at the end of training:

**NC1: Within-class variability collapse.** The features of samples from the same class converge to a mean feature vector. For any sample from class $c$, $f_{\boldsymbol{\theta}}(\boldsymbol{x}) \approx \boldsymbol{\mu}_c$ and $\Sigma_c \to \boldsymbol{0}$, where $\boldsymbol{\mu}_c = \frac{1}{n_c} \sum_{i=1}^{n_c} f_{\boldsymbol{\theta}}(\boldsymbol{x}_{c,i})$ is the mean feature of class $c$ and $\Sigma_c = \frac{1}{n_c} \sum_{i=1}^{n_c} (f_{\boldsymbol{\theta}}(\boldsymbol{x}_{c,i}) - \boldsymbol{\mu}_c)(f_{\boldsymbol{\theta}}(\boldsymbol{x}_{c,i}) - \boldsymbol{\mu}_c)^\top$ is the covariance.

**NC2: Simplex-ETF structure of class means.** Consider the global mean $\boldsymbol{\mu}_G = \frac{1}{C} \sum_{c=1}^{C} \boldsymbol{\mu}_c$. After mean-centering and normalization, the class means become equal-norm and equiangular:

$$\|\boldsymbol{\mu}_c - \boldsymbol{\mu}_G\|_2 - \|\boldsymbol{\mu}_{c'} - \boldsymbol{\mu}_G\|_2 \to 0, \quad \forall c, c' \in [C], \tag{3}$$

$$\langle \tilde{\boldsymbol{\mu}}_c, \tilde{\boldsymbol{\mu}}_{c'} \rangle \to \frac{C}{C-1} \delta_{c,c'} - \frac{1}{C-1}, \quad \forall c, c' \in [C], \tag{4}$$

where $\tilde{\boldsymbol{\mu}}_c = \frac{\boldsymbol{\mu}_c - \boldsymbol{\mu}_G}{\|\boldsymbol{\mu}_c - \boldsymbol{\mu}_G\|_2}$ and $\delta_{c,c'}$ is the Kronecker delta symbol (i.e., $\delta_{c,c'}$ equals to 1 when $c = c'$ and 0 otherwise).

**NC3: Self-duality between features and classifier.** The classifier weights $\boldsymbol{\phi}$ align with the class means $M = [\tilde{\boldsymbol{\mu}}_1, \tilde{\boldsymbol{\mu}}_2, \ldots, \tilde{\boldsymbol{\mu}}_C]$:

$$\left\| \frac{\boldsymbol{\phi}^\top}{\|\boldsymbol{\phi}\|_F} - \frac{\boldsymbol{M}}{\|\boldsymbol{M}\|_F} \right\|_F \to 0, \tag{5}$$

**NC4: Nearest-class-mean decision rule.** Because within-class scatter collapses and between-class means are symmetrically arranged, the linear classifier behaves as:

$$\arg\max_c (\langle \boldsymbol{a}_c, f_{\boldsymbol{\theta}}(\boldsymbol{x}) \rangle + b_c) \to \arg\min_c \|f_{\boldsymbol{\theta}}(\boldsymbol{x}) - \boldsymbol{\mu}_c\|_2, \tag{6}$$

where $\boldsymbol{a}_c$ and $b_c$ represent the weight and bias of the classifier for class $c$.

The NC observations motivate hard-wiring the last-layer classifier to the simplex-ETF geometry and training the feature extractor to adapt to it. Concretely, an ETF classifier is a linear head whose weight matrix $\boldsymbol{V} = [\boldsymbol{v}_1, \boldsymbol{v}_2, \ldots, \boldsymbol{v}_C] \in \mathbb{R}^{p \times C}$ is:

$$\boldsymbol{V} = \sqrt{\frac{C}{C-1}} \boldsymbol{U}(\boldsymbol{I}_C - \frac{1}{C} \mathbf{1}_C \mathbf{1}_C^\top), \tag{7}$$

where $p$ is the input dimension of ETF classifier, $\boldsymbol{U} \in \mathbb{R}^{p \times C}$ allows any rotation and satisfies $\boldsymbol{U}^\top \boldsymbol{U} = \boldsymbol{I}_C$, $\boldsymbol{I}_C$ is the identity matrix, and $\mathbf{1}_C$ is an all-ones vector.

## 4 METHOD

In this section, we introduce FedBlade, a federated learning framework integrating bilateral alignment and feature decorrelation. We present the full algorithm in Appendix B.

### 4.1 LDDECORR: ACCELERATE THE MITIGATION OF DIMENSIONAL COLLAPSE

**Linear gradients of FedDecorr.** To mitigate dimensional collapse caused by label skew, Shi et al. (2023) propose a regularization term named FedDecorr. This term regularizes the Frobenius norm of the representation correlation matrix during local training:

$$\mathcal{L}_{FedDecorr}(\boldsymbol{w}; \boldsymbol{X}) = \frac{1}{d^2} \|\boldsymbol{K}\|_F^2, \tag{8}$$

where $\boldsymbol{K}$ is the representation correlation matrix. This regularization term forces the correlation matrix to be full-rank, discouraging the tail singular values from collapsing to zero. However, as defined in Eq. (8), FedDecorr yields linear gradients and fails to guarantee the singular values $\lambda_i > 0$, since its penalty remains linear: $\nabla_{\lambda_i} = 2\lambda_i/d^2$.

To accelerate the mitigation of dimensional collapse, we revisit the regularization term. Given a correlation matrix $\boldsymbol{K}$, the dimensional collapse can be alleviated if $\boldsymbol{K}$ approaches the identity matrix $\boldsymbol{I}$. A key limitation of FedDecorr is that it treats all entries and implicitly all singular value deviations uniformly. Intuitively, stronger gradients should be applied to smaller singular values to more effectively prevent dimensional collapse. Motivated by this, we adopt the Log-Determinant (LogDet) divergence as the regularization term. The LogDet divergence is defined as follows.

**Definition 1 (LogDet Divergence).** Let $\mathcal{S}_+^d$ be the cone of $d \times d$ positive semi-definite (PSD) matrices. For $\boldsymbol{X}, \boldsymbol{Y} \in \mathcal{S}_+^d$, the LogDet divergence is defined as:

$$D_{ld}(\boldsymbol{X}, \boldsymbol{Y}) = \mathrm{tr}(\boldsymbol{X}\boldsymbol{Y}^{-1}) - \log\det(\boldsymbol{X}\boldsymbol{Y}^{-1}) - d. \tag{9}$$

To encourage $\boldsymbol{K}$ to approach the identity matrix $\boldsymbol{I}$, we minimize their LogDet divergence:

$$D_{ld}(\boldsymbol{K}, \boldsymbol{I}) = \mathrm{tr}(\boldsymbol{K}) - \log\det(\boldsymbol{K}) - d. \tag{10}$$

Since $\mathrm{tr}(\boldsymbol{K}) = d$, the LogDet divergence in Eq. (10) reduces to minimizing $-\log\det(\boldsymbol{K})$. We therefore formally define LDDecorr as a novel regularization term that minimizes the log-determinant of the representation correlation matrix during local training:

$$\mathcal{L}_{LDDecorr} = -\log\det(\boldsymbol{K}). \tag{11}$$

**Exponential gradients of LDDecorr.** With $\log\det(\boldsymbol{K}) = \sum_i \log \lambda_i$, LDDecorr yields exponential gradients: $\nabla_{\lambda_i} = -1/\lambda_i$. Unlike the linear gradients of FedDecorr, LDDecorr imposes an infinite penalty on small singular values, ensuring the correlation matrix remains full-rank and accelerating the mitigation of dimensional collapse. Experimental results in Sec. 5.3 validate the superiority of LDDecorr. Importantly, LDDecorr requires only determinant calculation, which is more efficient than calculating singular values. For further efficiency, we compute $\boldsymbol{K} = \boldsymbol{L}\boldsymbol{L}^\top$ via Cholesky factorization and evaluate $\log\det(\boldsymbol{K}) = 2\sum_i \log \boldsymbol{L}_{ii}$. Since $\boldsymbol{K}$ is PSD, we stabilize Cholesky factorization by replacing $\boldsymbol{K}$ with $\tilde{\boldsymbol{K}} = \boldsymbol{K} + \epsilon\boldsymbol{I}$, where $\epsilon = 10^{-4}$ serves as a small jitter. For a $d \times d$ symmetric positive definite matrix, calculating the determinant via Cholesky decomposition requires $\frac{1}{3}d^3$ FLOPs.

To quantify dimensional collapse, we measure the effective rank (Roy & Vetterli, 2007) of the representation covariance matrix, which reflects the effective dimensionality of the feature space. A higher effective rank indicates a lower degree of collapse. The effective rank is defined as follows.

**Definition 2 (Effective Rank).** For a matrix $\boldsymbol{A} \in \mathbb{R}^{m \times n}$ with non-zero singular values $\{\lambda_i\}_{i=1}^r$, define normalized weights $p_i = \lambda_i / \sum_{j=1}^r \lambda_j$, where $r = \min(m, n)$. The effective rank of $\boldsymbol{A}$ is defined as $eRank(\boldsymbol{A}) = \exp(\mathcal{H}(p_1, p_2, \ldots, p_r)) = \exp(-\sum_{i=1}^r p_i \log p_i)$, where $\mathcal{H}(\cdot)$ denotes the Shannon entropy.

By this definition, minimizing $\mathcal{L}_{LDDecorr}$ (equivalently, maximizing $\log \det(\boldsymbol{K})$) naturally increases the effective rank. Specifically, for the representation correlation matrix $\boldsymbol{K}$, the log-determinant $\sum_{i=1}^r \log \lambda_i$ is symmetric and concave, reaching its maximum when the spectrum is isotropic. Likewise, the Shannon entropy $\mathcal{H}(p_1, p_2, \ldots, p_r) = -\sum_{i=1}^r p_i \log p_i$ that defines the effective rank is also symmetric and concave, with the same maximum (i.e., isotropy). Thus, maximizing $\log \det \boldsymbol{K}$ pushes singular values away from zero and toward uniformity, increasing the effective rank and yielding $eRank(\boldsymbol{K}) = r$ at $\boldsymbol{K} = \boldsymbol{I}$.

### 4.2 PBA: PROTYTYPE-GUIDED BILATERAL ALIGNMENT

Another issue induced by label skew is classifier bias. To mitigate this, FedETF (Li et al., 2023) employs a fixed and synthetic ETF classifier shared across clients. We first introduce the supervised loss in FedETF. Specifically, a simplex ETF classifier $\boldsymbol{V} = [\boldsymbol{v}^1, \boldsymbol{v}^2, \ldots, \boldsymbol{v}^C] \in \mathbb{R}^{p \times C}$ is randomly initialized according to Eq.(7). Let $\boldsymbol{z}$ denote the feature vector and $f_{\boldsymbol{\Psi}}$ be the projector parameterized by $\boldsymbol{\Psi}$. The projector maps $\boldsymbol{z}$ into the ETF input space and normalize it to obtain the projected vector $\boldsymbol{\mu} = f_{\boldsymbol{\Psi}}(\boldsymbol{z})/\|f_{\boldsymbol{\Psi}}(\boldsymbol{z})\|_2$. Given the ETF classifier with weight matrix $\boldsymbol{V} = [\boldsymbol{v}^1, \boldsymbol{v}^2, \ldots, \boldsymbol{v}^C] \in \mathbb{R}^{p \times C}$, the supervised loss in FedETF is defined as:

$$\mathcal{L}_{sup}(\boldsymbol{\theta}, \boldsymbol{\Psi}, \boldsymbol{V}; \boldsymbol{x}, y) = -\log \frac{n_k^y \exp(\beta \cdot \boldsymbol{v}_y^\top \boldsymbol{\mu})}{\sum_{c \in [C]} n_k^c \exp(\beta \cdot \boldsymbol{v}_c^\top \boldsymbol{\mu})}, \tag{12}$$

where $n_k^c$ is the number of samples in class $c$ and $\beta$ is a learnable temperature. This loss is inspred by Balanced Softmax (Ren et al., 2020).

**The bridge for module synergy.** However, the synergy among the feature extractor, projector, and ETF classifier is overlooked. In FedETF, the projector becomes the last trainable layer under a fixed ETF classifier, which can be biased under label skew. Consequently, this layer may be misaligned with the classifier. To address this issue, we first analyze the roles of the feature extractor and projector. The feature extractor produces feature vectors for input samples, while the projector should map them close to the corresponding ETF classifier weights. Class prototypes, as the mean of feature vectors, provide natural bridges for aligning the projector with the ETF classifier, because projected prototypes should coincide with the shared ETF weights. For client $k$, each local class prototype $p_k^c \in \mathbb{R}^d$ is the mean feature vector within the same class:

$$\boldsymbol{p}_k^c = \frac{1}{n_k^c} \sum_{(\boldsymbol{x}, y) \in D_k^c} f_{\boldsymbol{\theta}_k}(\boldsymbol{x}), \quad \forall c \in [C], \tag{13}$$

where $D_k^c = \{(\boldsymbol{x}_i, y_i) \in D_k | y_c = c\}$ contains all samples assigned to class $c$. To provide a uniform input across clients, we calibrate the projector using global prototypes, which are aggregated as:

$$\bar{\boldsymbol{p}}^c = \sum_{k \in \mathcal{I}^{(t)}} \frac{n_k^c}{\sum_{i \in \mathcal{I}^{(t)}} n_i^c} \boldsymbol{p}_k^c, \quad \forall c \in [C]. \tag{14}$$

**Projector alignment.** Then, we introduce PBA, a prototype-guided bilateral alignment method that simultaneously aligns the feature extractor and projector via global prototypes. We first describe projector alignment. For each sample $(\boldsymbol{x}, c)$, the projected vector $\boldsymbol{\mu}$ should be close to the ETF classifier weight $\boldsymbol{v}^c$. As discussed above, each projected global prototype $\bar{\boldsymbol{\mu}}^c = f_{\boldsymbol{\Psi}}(\bar{\boldsymbol{p}}^c)/\|f_{\boldsymbol{\Psi}}(\bar{\boldsymbol{p}}^c)\|_2$ should also be close to corresponding ETF classifier weight. Motivated by this, we introduce a loss term to measure the cosine distance between the projected global prototypes and corresponding ETF classifiers:

$$\mathcal{L}_{PA} = \sum_{c \in [C]} \frac{1}{2} \left(1 - \bar{\boldsymbol{\mu}}_c^\top \boldsymbol{v}^c\right)^2, \tag{15}$$

where $\bar{\boldsymbol{\mu}}^c$ and $\boldsymbol{v}^c$ are $l_2$ normalized global prototypes and classifier weights, respectively. This loss term calibrates the projector, enabling the synergy between the projector and ETF classifier.

**Feature extractor alignment.** Moreover, to enhance the consistency of the feature extractor, we simultaneously align it with global prototypes. However, similar to CrossEntropy loss, conventional contrastive alignment can be biased under label skew. Inspired by Balanced Softmax (Ren et al., 2020), we incorporate class distributions to balance gradients. The balanced feature alignment loss is defined as:

$$\mathcal{L}_{FA} = -\log \frac{n_k^c \exp(sim(f_{\boldsymbol{\theta}}(\boldsymbol{x}), \bar{\boldsymbol{p}}^c)/\tau)}{\sum_{i=1}^{C} n_k^i \exp(sim(f_{\boldsymbol{\theta}}(\boldsymbol{x}), \bar{\boldsymbol{p}}^i)/\tau)}, \tag{16}$$

where $sim(\boldsymbol{a}, \boldsymbol{b})$ denotes cosine similarity and $\tau$ is a temperature parameter. By combining projector alignment $\mathcal{L}_{PA}$ and feature alignment $\mathcal{L}_{FA}$, our PBA enforces the synergy among the feature extracor, projector and ETF classifier.

**Local objective of FedBlade.** Finally, by integrating LDDecorr and PBA, the local objective of FedBlade can be formulated as:

$$\mathcal{L} = \mathcal{L}_{sup} + \beta \cdot \mathcal{L}_{LDDecorr} + \gamma \cdot (\mathcal{L}_{PA} + \mathcal{L}_{FA}), \tag{17}$$

where $\beta$ controls the strength of feature decorrelation and $\gamma$ is the weight of prototype-guided bilateral alignment. Two component address distinct yet interdependent challenges under label skew, and their integration is essential for achieving strong performance, as demonstrated by the ablation results in Tab. 3.

# 5 EXPERIMENTS

## 5.1 EXPERIMENTAL SETUPS

**Datasets.** We consider three classical datasets, including CIFAR-10/CIFAR-100 (Krizhevsky et al., 2009) and Tiny-ImageNet (Le & Yang, 2015). Following prior works (Wang et al., 2020a; Li et al., 2021; Shi et al., 2023), we adopt a common label skew setting in federated learning, namely Dirichlet distribution $Dir(\alpha)$. The argument $\alpha$ controls the level of label skew, where smaller $\alpha$ means more severe skew. We conduct our experiments on three Dirichlet distributions: $Dir(0.05)$, $Dir(0.1)$ and $Dir(0.5)$.

**Baselines.** We compare FedBlade with several federated learning methods that address label skew, falling under the following categories: (1) classical FL methods: FedAvg (McMahan et al., 2017) and FedProx (Li et al., 2020); (2) Logits calibration: FedLC (Zhang et al., 2022); (3) Feature alignment: FedProto (Tan et al., 2022) and FedFM (Ye et al., 2023a); (4) Dimensional collapse mitigation: FedDecorr (Shi et al., 2023) and FedRCL (Seo et al., 2024); and (5) Fixed ETF classifier: FedETF (Li et al., 2023).

**Implementation details.** For all three datasets, we evaluate under two FL settings: (1) partial participation, where 20 clients are randomly sampled from 100 at each round and communication round is 200; and (2) full participation, where all 20 clients participate at each round and communication round is 100. For all datasets, we use MobileNetV2 (Sandler et al., 2018). Local training is performed for 5 epochs using SGD optimizer with a learning rate of 0.01, a momentum of 0.9, and a weight decay of 0.00001. The batch size is 64. $\beta$ and $\gamma$ in Eq.(17) are set to 0.005 and 1, respectively. Each experiment is repeated three times with different random seeds {1024, 2025, 4096}, and we report the averaged accuracy over the last 10 rounds. Additional hyperparameter details are provided in Appendix D.

## 5.2 MAIN RESULTS

**Test accuracy.** We first evaluate on three datasets under the partial participation setting. We report the averaged accuracy over the last 10 rounds in Tab. 1. The results show that FedBlade consistently outperforms existing methods. In particular, FedBlade provides modest improvements on CIFAR-10 but achieves substantially larger gains on CIFAR-100 and Tiny-ImageNet. This is because that

decision boundaries become geometrically narrower as the number of classes $C$ increases, making classification more sensitive to feature bias. By mitigating dimensional collapse and aligning the projector with the ETF classifier, FedBlade produces wider decision margins among confusable classes. We also conduct experiments under the full participation setting, with results reported in Appendix E.1.

Table 1: **Accuracy (%) comparisons under the partial partition.** 20 clients are selected from 100 clients per round. All results are averaged over 3 runs (mean $\pm$ std). The best and second results are highlighted with bold and underline, respectively.

| Method | CIFAR-10 | | | CIFAR-100 | | | Tiny-ImageNet | | |
|---|---|---|---|---|---|---|---|---|---|
| | $Dir(0.05)$ | $Dir(0.1)$ | $Dir(0.5)$ | $Dir(0.05)$ | $Dir(0.1)$ | $Dir(0.5)$ | $Dir(0.05)$ | $Dir(0.1)$ | $Dir(0.5)$ |
| FedAvg | 55.19±3.81 | 69.26±2.56 | 86.12±0.32 | 50.70±0.46 | 55.52±0.41 | 60.40±0.21 | 33.72±0.52 | 37.68±0.33 | 41.59±0.28 |
| FedProx | 53.74±5.13 | 69.53±3.01 | 85.94±0.40 | 50.97±0.43 | 55.33±0.36 | 60.41±0.15 | 33.15±0.56 | 37.29±0.36 | 41.64±0.26 |
| FedLC | 75.10±0.90 | 80.71±0.22 | 86.71±0.12 | 51.12±0.26 | 55.35±0.30 | 60.30±0.19 | 37.09±0.26 | 40.12±0.12 | 42.42±0.25 |
| FedDecorr | 57.77±3.51 | 70.85±2.95 | 85.78±0.27 | 50.86±0.26 | 54.26±0.35 | 58.87±0.14 | 35.87±0.51 | 38.77±0.38 | 41.82±0.19 |
| FedRCL | 52.14±3.71 | 71.15±1.57 | 86.91±0.23 | 50.56±0.24 | 56.71±0.35 | 61.32±0.16 | 31.94±0.54 | 36.81±0.40 | 41.95±0.29 |
| FedProto | 55.05±4.11 | 69.35±2.86 | 85.96±0.30 | 50.95±0.46 | 55.81±0.42 | 60.64±0.21 | 31.31±0.49 | 36.47±0.46 | 42.45±0.28 |
| FedFM | 55.04±4.09 | 69.61±2.84 | 86.52±0.53 | 46.55±0.57 | 54.83±0.55 | 61.98±0.26 | 25.41±0.80 | 33.56±0.45 | 40.47±0.37 |
| FedETF | 75.80±0.46 | 80.66±0.32 | 86.56±0.08 | 51.41±1.18 | 55.31±0.26 | 58.81±1.84 | 37.09±0.29 | 40.03±0.14 | 41.96±0.29 |
| FedBlade | **75.83±0.70** | **81.67±0.30** | **87.90±0.14** | **54.31±0.19** | **57.92±0.15** | **62.07±0.17** | **39.43±0.17** | **41.88±0.15** | **43.63±0.25** |

Table 2: **Convergence speed under $Dir(0.05)$. Left:** CIFAR-100. **Right:** Tiny-ImageNet. 20 clients are selected from 100 clients per round. FedBlade significantly speeds up the convergence of the global model.

| Method | 40% accuracy | | 50% accuracy | | | Method | 20% accuracy | | 30% accuracy | |
|---|---|---|---|---|---|---|---|---|---|---|
| | #Rounds | Speedup | #Rounds | Speedup | | | #Rounds | Speedup | #Rounds | Speedup |
| FedAvg | 85 | $(1.0\times)$ | 184 | $(1.0\times)$ | | FedAvg | 70 | $(1.0\times)$ | 141 | $(1.0\times)$ |
| FedBlade | **48** | **$(1.7\times)$** | **105** | **$(1.9\times)$** | | FedBlade | **41** | **$(1.7\times)$** | **81** | **$(1.7\times)$** |
| FedDecorr | 68 | $(1.3\times)$ | 176 | $(1.0\times)$ | | FedDecorr | 45 | $(1.6\times)$ | 101 | $(1.4\times)$ |
| FedETF | 74 | $(1.1\times)$ | 151 | $(1.2\times)$ | | FedETF | 60 | $(1.2\times)$ | 116 | $(1.2\times)$ |

**Convergence speed.** Tab. 2 reports the communication round at which each representative method first reaches the specified accuracy. Benefiting from LDDecorr and module synergy, FedBlade achieves substantially faster convergence. Additional results are provided in Appendix E.8.

## 5.3 ABLATION STUDY

**Key components.** To assess the effectiveness of the two key components in FedBlade, we conduct an ablation study on CIFAR-100 and Tiny-ImageNet under the partial participation setting. Tab. 3 reports the results across different levels of label skew. Notably, removing both LDDecorr and PBA degenerates FedBlade into FedETF (i.e., the first row of Tab. 3). We observe that both components are essential. Removing either leads to performance degradation, in some cases even worse than vanilla FedETF. The synergy emerges because each module provides what the other lacks. LDDecorr ensures that the representation space retains enough dimensionality for PBA to generate meaningful prototypes, while PBA imposes structured geometry that counteracts the potential instability caused by strong decorrelation.

Table 3: **Ablation study on key components.** 20 clients are selected from 100 clients per round. The first row is vanilla FedETF. Both components are essential for FedBLADE.

| LDDecorr | PBA | CIFAR-100 | | | Tiny-ImageNet | | |
|---|---|---|---|---|---|---|---|
| | | $Dir(0.05)$ | $Dir(0.1)$ | $Dir(0.5)$ | $Dir(0.05)$ | $Dir(0.1)$ | $Dir(0.5)$ |
| | | 51.41±1.18 | 55.31±0.26 | 58.81±1.84 | 37.09±0.29 | 40.03±0.14 | 41.96±0.29 |
| ✓ | | 50.98±0.20 | 53.74±0.13 | 56.24±0.17 | 39.20±0.15 | 40.46±0.20 | 37.43±0.27 |
| | ✓ | 52.66±0.50 | 56.42±0.16 | 61.60±0.19 | 36.05±0.48 | 39.55±0.21 | 41.82±0.46 |
| ✓ | ✓ | **54.31±0.19** | **57.92±0.15** | **62.07±0.17** | **39.43±0.17** | **41.88±0.15** | **43.63±0.25** |

**Feature decorrelation.** We evaluate the effectiveness of LDDecorr through an ablation study on feature decorrelation methods. Fig. 2 shows that LDDecorr more effectively prevents tail singular values from collapsing to zero, suggesting that LDDecorr imposes an infinite penalty on small singular values (as discussed in Sec. 4.1). Fig. 3 further shows that Fed-Blade with LDDecorr achieves faster convergence and higher test accuracy than FedBlade with FedDecorr. Besides, FedBlade with either feature decorrelation method consistently outperforms FedETF and FedAvg. To quantify the mitigation of dimensional collapse, we plot the effective rank of the representation correlation

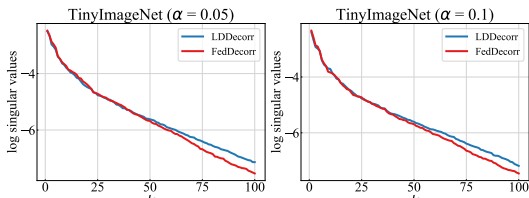

Figure 2: **Effects of LDDecorr on mitigating dimensional collapse.** We plot the singular values of the representation covariance matrix. The $x$-axis indicates the indices of the singular values and the $y$-axis is the logarithm of singular values. LDDecorr effectively prevents the tail singular values from collapsing to zero.

matrix over communication rounds in Fig. 4. In particular, we evaluate the effective rank in the output space of projector for FedETF and FedBlade. This can measure the final embedding space used by the classifier. As expected, feature decorrelation increases the effective rank. Furthermore, FedBlade with LDDecorr provides stronger mitigation, which is indicated by higher effective rank. These observations verify that (1) mitigating dimensional collapse speeds up global model convergence, and (2) LDDecorr further accelerates this mitigation by imposing infinite penalty on small singular values.

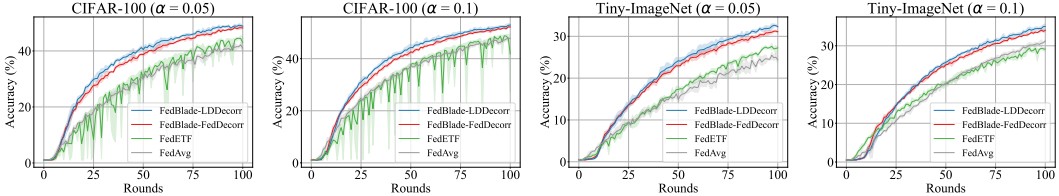

Figure 3: **Test accuracy (%) under various label skew settings on CIFAR-100 and Tiny-ImageNet.** FedBlade with LDDecorr achieves faster convergence speed.

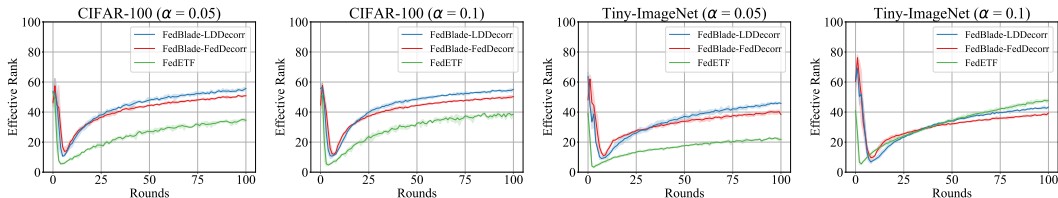

Figure 4: **Effective rank under various label skew settings on CIFAR-100 and Tiny-ImageNet.** The effective rank is computed in the output space of projector. FedBlade with LDDecorr achieves higher effective rank.

To evaluate the effective rank of all methods, we compute this metric in the output space of feature extractor that is shared across architectures. The results on CIFAR-100 ($\alpha = 0.05$) and Tiny-ImageNet ($\alpha = 0.05$) are shown in Fig. 5, and more results are provided in Appendix E.4. These results indicate that *effective rank and accuracy are not strictly monotonic*. Once feature diversity is sufficient, excessive rank expansion can degrade class structure. This phenomenon can be supported by the observations in CW-RGP (Weng et al., 2022), where over-whitened features can break the potential manifold the examples in the same class belong to, making the learning more difficult. Besides, appropriate effective rank can be also supported by neural collapse (Papyan et al., 2020), where good generalization is associated with structured high-dimensional geometries (i.e., simplex ETF), rather than arbitrarily increasing the dimensionality of representations. Combining LDDecorr with PBA yields both higher effective rank and a more structured ETF-like geometry, explaining why FedBlade achieves stronger performance.

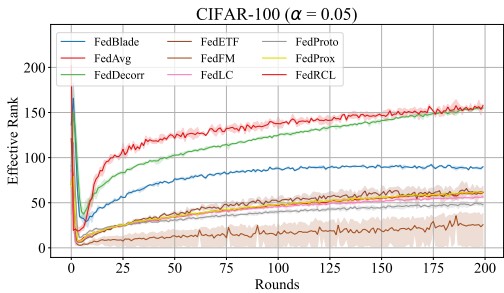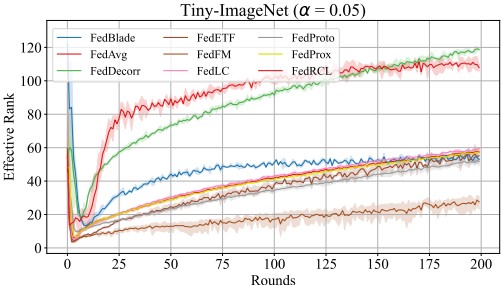

Figure 5: Effective rank on CIFAR-100 ($\alpha = 0.05$) and Tiny-ImageNet ($\alpha = 0.05$). The effective rank is computed in the output space of feature extractor.

**Bilateral alignment.** We ablate the two alignment terms in PBA, namely $\mathcal{L}_{FA}$ and $\mathcal{L}_{PA}$. Tab. 4 shows that both terms are essential. Removing projector alignment (PA) misaligns the projector with the ETF classifier, leading to consistent degradation across all label-skew levels. Excluding feature alignment (FA) reduces performance under $Dir(0.5)$, where FA is more effective. Moreover, as discussed in Sec. 4.2, incorporating class distributions balances the gradients; thus, removing distribution factor (DF) in Eq.(16) causes significant performance drops under severe skew.

**Quantitative analysis of PBA.** To demonstrate that PBA encourages neural collapse, we quantify two standard metrics on CIFAR-100: within-class variance (**NC1**) and deviation from the simplex ETF structure (**NC2**), where lower values indicate stronger neural-collapse behavior. As shown in Tab. 5, adding PBA consistently reduces both **NC1** and **NC2** across all heterogeneity settings, demonstrating that PBA promotes tighter class clusters and more ETF-like feature geometry. These results provide direct quantitative evidence that PBA contributes to the formation of neural collapse.

Table 4: **Ablation study on two loss terms of PBA.** "w/o FA" means removing $\mathcal{L}_{FA}$ in Eq.(17), "w/o DF" means removing the distribution factor in Eq.(16), and "w/o PA" means removing $\mathcal{L}_{PA}$ in Eq.(17).

| $\alpha =$ | 0.05 | 0.1 | 0.5 |
|---|---|---|---|
| w/o FA | 39.36±0.18 | 41.19±0.11 | 42.11±0.22 |
| w/o DF | 36.61±0.56 | 39.95±0.33 | 43.40±0.23 |
| w/o PA | 38.68±0.27 | 40.48±0.20 | 42.50±0.28 |
| FedBlade | **39.43±0.17** | **41.88±0.15** | **43.63±0.25** |

Table 5: **The efficacy of PBA on enforcing neural collapse.** ↓ means that a lower value is better.

| $Dir(\alpha)$ | Method | NC1 ↓ | NC2 ↓ |
|---|---|---|---|
| $Dir(0.05)$ | w/o PBA | 0.6425 | 22.3306 |
| | w/ PBA | 0.5669 | 18.5304 |
| $Dir(0.1)$ | w/o PBA | 0.6367 | 21.4161 |
| | w/ PBA | 0.5268 | 17.0571 |
| $Dir(0.5)$ | w/o PBA | 0.6260 | 20.8275 |
| | w/ PBA | 0.4762 | 16.3720 |

## 6 CONCLUSION

In this paper, we take a further step toward label skew in federated learning. We have presented **FedBlade**, a **fed**erated learning framework with **b**ilateral **a**lignment and feature **de**correlation. Experimental results show that our feature decorrelation method prevents the small singular values from collapsing to zero, further mitigating dimensional collapse. Besides, when fixing ETF classifier across clients, our bilateral alignment method promotes the synergy among the feature extractor, projector and ETF classifier. The two components address *distinct yet interdependent challenges under label skew*. Although feature decorrelation effectively mitigates dimensional collapse, this method is sensitive to the decorrelation strength. We will investigate other regularization methods to address dimensional collapse in the future. We hope that FedBlade can inspire more studies on the mitigation of dimensional collapse and FL methods with fixed ETF classifier.

REPRODUCIBILITY STATEMENT

We present the details of our method in Sec. 4 and Algorithm 1. We provide the details of experimental setups in Sec. 5.1 and Appendix D. The calculation of experimental metrics is described in Sec. 5.1. We will provide our code during the rebuttal phase upon request, and release it publicly upon acceptance.

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

## A  TABLE OF NOTATIONS

Please refer to Tab. 6 for the notations used throughout this paper.

| Notation | Description |
|----------|-------------|
| $\mathcal{X}$ | Input space |
| $\mathcal{Z}$ | Feature space |
| $\mathcal{Y}$ | Label space |
| $\mathcal{L}$ | Loss function |
| $K$ | Number of all clients |
| $D_k$ | Training dataset of client $k$ |
| $C$ | Number of all classes |
| $n_k$ | Size of dataset $D_k$ |
| $n_k^c$ | Number of samples from class $c$ in dataset $D_k$ |
| $f_{\boldsymbol{\theta}}$ | Feature extractor parameterized by $\boldsymbol{\theta}$ |
| $f_{\boldsymbol{\Psi}}$ | Projector parameterized by $\boldsymbol{\Psi}$ |
| $f_{\boldsymbol{\phi}}$ | Classifier parameterized by $\boldsymbol{\phi}$ |
| $\boldsymbol{x}$ | Input |
| $\boldsymbol{z}$ | Feature vector generated by $f_{\boldsymbol{\theta}}$ |
| $y$ | Label |
| $d$ | Dimensionality of feature space |
| $p$ | Dimensionality of projection space |
| $\mathcal{I}^{(t)}$ | Selected clients at round $t$ |
| $\boldsymbol{w}$ | Global model |
| $\boldsymbol{w}_k$ | Local model of client $k$ |
| $\sum$ | Representation covariance matrix |
| $\boldsymbol{K}$ | Representation correlation matrix |
| $\boldsymbol{V}$ | Weight matrix of ETF classifier |
| $\lambda_i$ | $i$-th singular value |
| $\boldsymbol{p}_k^c$ | Client $k$' local prototype of class $c$ |
| $\bar{\boldsymbol{p}}^c$ | Global prototype of class $c$ |

Table 6: Table of notations.

## B  ALGORITHM

The procedure of FedBlade is formally presented in Algorithm 1.

## C  DETAILS OF DATASETS

We first ourline the details of the datasets used in our experiments.

• CIFAR-10 dataset contains 60,000 color samples with size of 32*32 pixels. This dataset is divided into 10 distinct classes and split into 50,000 training and 10,000 test samples. Each class contains 6,000 samples.

• CIFAR-100 dataset builds on CIFAR-10 by increasing the number of classes from 10 to 100, while keeping the same image size of 32×32 pixels. It contains the same total number of samples, i.e., 60,000 samples, but with only 600 samples per class. For each class, 500 samples are used for training and 100 samples are used to testing.

• Tiny-ImageNet dataset is a scaled-down version of the larger ImageNet dataset. This dataset is designed to provide a middle ground between small datasets like CIFAR and the massive ImageNet dataset. This dataset contains 200 classes, each with 500 training samples and 50 test samples. The total size is 120,000. The image size is 64×64 pixels.

---

**Algorithm 1** FedBlade

---

1: **Input:** number of communication rounds $T$, initial model $\boldsymbol{w}$, local epochs $E$, learning rate $\eta$, feature decorrelation strength $\beta$, and bilateral alignment weight $\gamma$.
2: **for** $t = 0, 1, \ldots, T - 1$ **do**
3:     // Server executes:
4:     Send global model $\boldsymbol{w}^{(t)}$ to each client
5:     Send global prototypes $\{\bar{\boldsymbol{p}}_c^{(t)}\}_{c \in [C]}$ to each client
6:     // Client executes:
7:     **for** each client $k \in \mathcal{I}^{(t)}$ in parallel **do**
8:         Set $\boldsymbol{w}_k^{(t)} = \boldsymbol{w}^{(t)}$
9:         **for** epoch $e = 1, 2, \ldots, E$ **do**
10:             **for** each mini-batch $\mathcal{B}$ **do**
11:                 Compute supervised loss $\mathcal{L}_{sup}$ by Eq. (12)
12:                 Compute feature decorrelation loss $\mathcal{L}_{LDDecorr}$ by Eq. (11)
13:                 Compute projector alignment loss $\mathcal{L}_{PA}$ by Eq. (15)
14:                 Compute feature alignment loss $\mathcal{L}_{FA}$ by Eq. (16)
15:                 $\mathcal{L} = \mathcal{L}_{sup} + \beta \cdot \mathcal{L}_{LDDecorr} + \gamma \cdot (\mathcal{L}_{PA} + \mathcal{L}_{FA})$
16:                 $\boldsymbol{w}_k^{(t)} \leftarrow \boldsymbol{w}_k^{(t)} - \eta \nabla \mathcal{L}(\boldsymbol{w}_k^{(t)}; \mathcal{B})$
17:             **end for**
18:         **end for**
19:         **for** $c \in [C]$ **do**
20:             Generate local prototype $\boldsymbol{p}_{k,c}^{(t)}$ by Eq. (13)
21:         **end for**
22:         Send $\boldsymbol{w}_k^{(t)}$ and $\{\boldsymbol{p}_{k,c}^{(t)}\}_{c \in [C]}$ to server
23:     **end for**
24:     // Server executes:
25:     Update global model $\boldsymbol{w}^{(t+1)}$ by Eq. (2)
26:     **for** each class $c \in [C]$ **do**
27:         Update global prototype $\bar{\boldsymbol{p}}_c^{(t+1)}$ by Eq. (14)
28:     **end for**
29: **end for**

---

Then, we introduce the data augmentation used in our experiments. For all three datasets, we follow the standard data augmentation and normalization process. Specifically, we first use Random-Crop(32, padding=4) and RandomHorizontalFlip(). Then, for CIFAR-10 and CIFAR-100, each channels (r, g, b) are normalized by mean $\mu = (0.4914, 0.4822, 0.4465)$ and standard deviation $\sigma = (0.2023, 0.1994, 0.2010)$, respectively. For Tiny-ImageNet, each channels are normalized by mean $\mu = (0.47889522, 0.47227842, 0.43047404)$ and standard deviation $\sigma = (0.24205776, 0.23828046, 0.25874835)$. For test dataset, we only perform the normalization process.

## D  DETAILS OF EXPERIMENTAL SETUPS

All experiments were conducted on a server equipped with two NVIDIA RTX 4090 GPUs, an AMD Ryzen 9 9950X CPU, and 128 GB of RAM. All results were produced using PyTorch 2.6.0, under Ubuntu 22.04.

For all three datasets under partial participation and full participation, we use MobileNetV2 (Sandler et al., 2018) and adopt SGD as the optimizer. For all methods, the learning rate is set to 0.01, the momentum is set to 0.9, the weight decay is set to 0.00001, the local epoch is set to 5, and the batch size is set to 64. For partial participation setting, the communication round is set to 200; for full participation setting, the communication round is set to 100.

For FedBlade, we turn the feature decorrelation strength $\beta \in \{0.001, 0.005, 0.01, 0.05\}$, and set it to 0.005 according to the sensitivity analysis in Fig. 6. We turn the bilateral alignment weight $\gamma \in \{0.1, 1.0, 2.0, 5.0\}$, and set it to 1.0 according to the sensitivity analysis in Fig. 7. We turn the temperature $\tau \in \{0.01, 0.05, 0.1, 0.5\}$, and set it to 0.1 according to the sensitivity analysis in Fig. 8.

Here, we list the hyperparameters for all baselines.

- FedProx (Li et al., 2020): regularization weight $\mu$ is set to 0.01.

- FedLC (Zhang et al., 2022): constant $\tau$ in the logits calibration is set to 10.

- FedDecorr (Shi et al., 2023): feature decorrelation weight $\beta$ is set to 0.1.

- FedRCL (Seo et al., 2024): regularization weight $\beta$ is set to 0.7, and temperature $\tau$ is set to 0.1.

- FedProto (Tan et al., 2022): alignment weight $\lambda$ is set to 1.

- FedFM (Ye et al., 2023a): alignment weight $\lambda$ is set to 1, and temperature $\tau$ is set to 0.1.

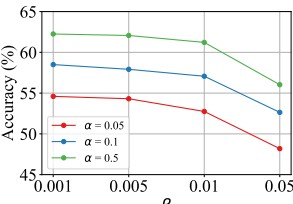 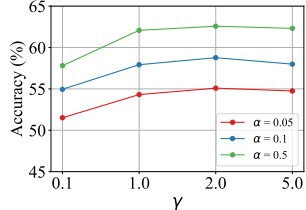 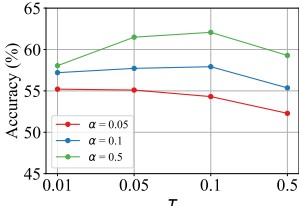

Figure 6: Sensitivity analysis of feature decorrelation strength $\beta$. Figure 7: Sensitivity analysis of bilateral alignment weight $\gamma$. Figure 8: Sensitivity analysis of temperature parameter $\tau$.

# E  ADDITIONAL EXPERIMENTAL RESULTS

## E.1  TEST ACCURACY UNDER FULL PARTICIPATION

We evaluate on three datasets under the full participation setting. Tab. 7 reports the averaged test accuracy over the last 10 rounds. Under full client participation, each client has access to more local data and participates in every round, which reduces data heterogeneity and stabilizes prototype generation. In this setting, the challenges that FedBlade is designed to address, i.e., dimensional collapse and feature inconsistency under label skew, are less pronounced. However, we note that FedBlade remains competitive and does not degrade performance in this setting.

Table 7: **Accuracy (%) comparisons under the full partition.** The model is MobileNetV2. All 20 clients are selected per round. All results are averaged over 3 runs (mean ± std). The best and second results are highlighted with bold and underline, respectively.

| Method | CIFAR-10 | | | CIFAR-100 | | | Tiny-ImageNet | | |
|---|---|---|---|---|---|---|---|---|---|
| | $Dir(0.05)$ | $Dir(0.1)$ | $Dir(0.5)$ | $Dir(0.05)$ | $Dir(0.1)$ | $Dir(0.5)$ | $Dir(0.05)$ | $Dir(0.1)$ | $Dir(0.5)$ |
| FedAvg | 67.44±1.02 | 81.88±0.45 | 89.86±0.06 | 57.94±0.26 | 61.80±0.19 | 66.82±0.11 | 43.02±0.44 | 46.35±0.31 | 51.24±0.31 |
| FedProx | 70.21±1.01 | 82.99±0.16 | 89.62±0.06 | 57.48±0.20 | 61.87±0.17 | 66.35±0.20 | 42.13±0.34 | 45.35±0.30 | 50.01±0.28 |
| FedLC | 75.09±0.13 | **84.81±0.13** | 89.78±0.10 | 56.63±0.11 | 61.17±0.16 | 66.42±0.10 | 44.73±0.24 | 47.66±0.33 | 51.08±0.18 |
| FedDecorr | 73.55±0.48 | 84.07±0.11 | 89.35±0.11 | 56.56±0.11 | 60.59±0.10 | 65.09±0.11 | 44.34±0.26 | 46.63±0.24 | 50.96±0.25 |
| FedRCL | 61.25±0.35 | 76.67±0.28 | 89.85±0.12 | 53.07±0.20 | 60.37±0.16 | 67.10±0.17 | 38.17±0.44 | 42.86±0.40 | 48.88±0.34 |
| FedProto | 70.67±0.31 | 83.59±0.10 | 89.76±0.08 | 57.17±0.21 | 61.89±0.13 | 66.48±0.16 | 41.05±0.36 | 45.16±0.32 | 51.18±0.37 |
| FedFM | 66.15±0.54 | 82.85±1.11 | 90.20±0.08 | 52.36±4.17 | 62.47±0.19 | 67.63±0.16 | 37.72±0.50 | 42.85±0.51 | 48.65±0.37 |
| FedETF | 75.22±0.23 | 84.66±0.15 | 89.66±0.12 | 57.49±0.30 | 61.77±0.26 | 66.65±0.14 | **45.50±0.40** | **48.59±0.35** | 51.71±0.35 |
| FedBlade | **76.25±0.20** | 84.20±0.12 | **90.44±0.06** | **58.33±0.24** | **62.57±0.19** | **68.13±0.08** | 43.99±0.28 | 47.69±0.32 | **51.99±0.30** |

## E.2  TEST ACCURACY ON RESNET-18

To evaluate our method on different model architectures, we conduct experiments on ResNet-18. The feature dimension of ResNet-18 is 512, which is smaller than that of MobileNetV2 with 1280

Table 8: **Accuracy (%) comparisons under the partial partition.** The model is ResNet-18. 20 clients are selected from 100 clients per round. All results are averaged over 3 runs (mean $\pm$ std). The best and second results are highlighted with bold and underline, respectively.

| Method | CIFAR-10 | | | CIFAR-100 | | | Tiny-ImageNet | | |
|---|---|---|---|---|---|---|---|---|---|
| | $Dir(0.05)$ | $Dir(0.1)$ | $Dir(0.5)$ | $Dir(0.05)$ | $Dir(0.1)$ | $Dir(0.5)$ | $Dir(0.05)$ | $Dir(0.1)$ | $Dir(0.5)$ |
| FedAvg | 62.09$\pm$3.67 | 73.78$\pm$4.11 | 88.69$\pm$0.57 | 57.08$\pm$0.47 | 60.81$\pm$0.44 | 63.44$\pm$0.25 | 39.70$\pm$0.48 | 43.29$\pm$0.25 | 46.29$\pm$0.20 |
| FedProx | 62.76$\pm$3.97 | 74.68$\pm$3.51 | 88.65$\pm$0.55 | 56.97$\pm$0.33 | 60.79$\pm$0.38 | 63.19$\pm$0.25 | 39.84$\pm$0.40 | 43.45$\pm$0.68 | 46.38$\pm$0.32 |
| FedLC | 79.83$\pm$0.62 | 84.68$\pm$0.31 | 89.20$\pm$0.17 | 56.62$\pm$0.64 | 58.77$\pm$0.27 | 61.93$\pm$0.20 | 42.65$\pm$0.17 | 45.66$\pm$0.17 | 47.37$\pm$0.21 |
| FedDecorr | 67.05$\pm$3.33 | 76.53$\pm$3.91 | 88.74$\pm$0.41 | 54.72$\pm$0.38 | 57.40$\pm$0.35 | 58.14$\pm$0.32 | 40.17$\pm$0.46 | 42.92$\pm$0.27 | 44.83$\pm$0.19 |
| FedRCL | 63.83$\pm$3.02 | 71.07$\pm$1.81 | 86.84$\pm$0.34 | 52.42$\pm$0.31 | 58.70$\pm$0.38 | 63.49$\pm$0.26 | 37.00$\pm$0.33 | 41.61$\pm$0.40 | 45.53$\pm$0.33 |
| FedProto | 61.58$\pm$3.03 | 74.41$\pm$4.08 | 89.16$\pm$0.47 | 56.18$\pm$0.71 | 59.69$\pm$0.43 | 64.27$\pm$0.25 | 39.43$\pm$0.72 | 43.53$\pm$0.33 | 47.82$\pm$0.24 |
| FedFM | 63.79$\pm$2.83 | 75.48$\pm$4.73 | 88.88$\pm$0.85 | 55.43$\pm$0.56 | 61.42$\pm$0.68 | **67.40$\pm$0.24** | 34.12$\pm$1.25 | 41.54$\pm$0.71 | **48.08$\pm$0.29** |
| FedETF | 80.41$\pm$0.61 | 85.08$\pm$0.38 | 89.36$\pm$0.13 | 57.61$\pm$0.70 | 60.09$\pm$1.21 | 60.92$\pm$1.23 | 42.79$\pm$0.56 | 45.41$\pm$0.18 | 47.08$\pm$0.52 |
| FedBlade | **81.41$\pm$0.50** | **85.66$\pm$0.13** | **90.25$\pm$0.13** | **60.56$\pm$0.20** | **63.94$\pm$0.27** | 65.55$\pm$0.13 | **43.43$\pm$0.28** | **45.84$\pm$0.26** | 47.94$\pm$0.25 |

dimensions. Tab. 8 reports the averaged test accuracy over the last 10 rounds under partial participation setting. We find that FedBLADE can still achieve strong performance on ResNet-18.

### E.3 CONVERGENCE CURVES

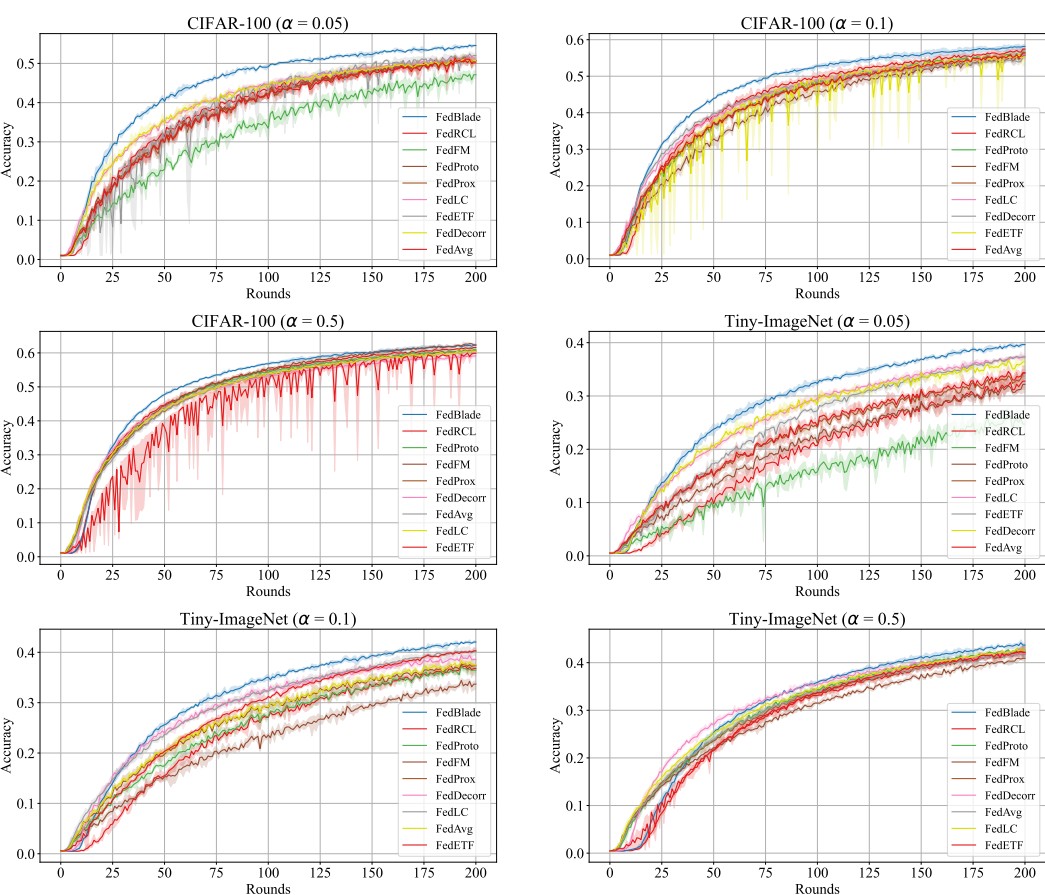

Figure 9: **Test accuracy under various label skew settings on CIFAR-100 and Tiny-ImageNet.** FedBLADE achieves faster convergence speed compared with other baselines, especially under severe label skew (e.g., $Dir(0.05)$).

As discussed in Sec. 4.1, feature decorrelation helps mitigate dimensional collapse during local training, thereby accelerating the convergence of the global model. Besides, as stated in Sec. 4.2 the synergy among the feature extractor, projector and ETF classifier can further improve the per-

formance of the global model. To compare the performance of different FL methods, we plot the accuracy curve over communication rounds under partial participation setting. Fig. 9 shows the experimental results on CIFAR-100 and Tiny-ImageNet under various label skew settings, including $Dir(0.05)$, $Dir(0.1)$ and $Dir(0.5)$. The results illustrate that FedBlade achieves substantially faster convergence under the above settings, indicating the effectiveness of our LDDecorr and PBA.

### E.4 ADDITIONAL EFFECTIVE RANK RESULTS

We visualize the effective rank for all methods in Fig. 10. In particular, the effective rank of all methods in Fig. 10 is computed in the output space of feature extractor, which is shared across architectures. As discussed in 5.3, FedBlade increases effective rank relative to some baselines, but these results indicate that effective rank and accuracy are not strictly monotonic. By pulling representations toward their corresponding ETF directions, PBA provides a structured geometric anchor that counteracts the potential side effects of strong feature decorrelation, while preserving the intended rank-expansion benefits of LDDecorr. Therefore, FedBlade achieves higher performance.

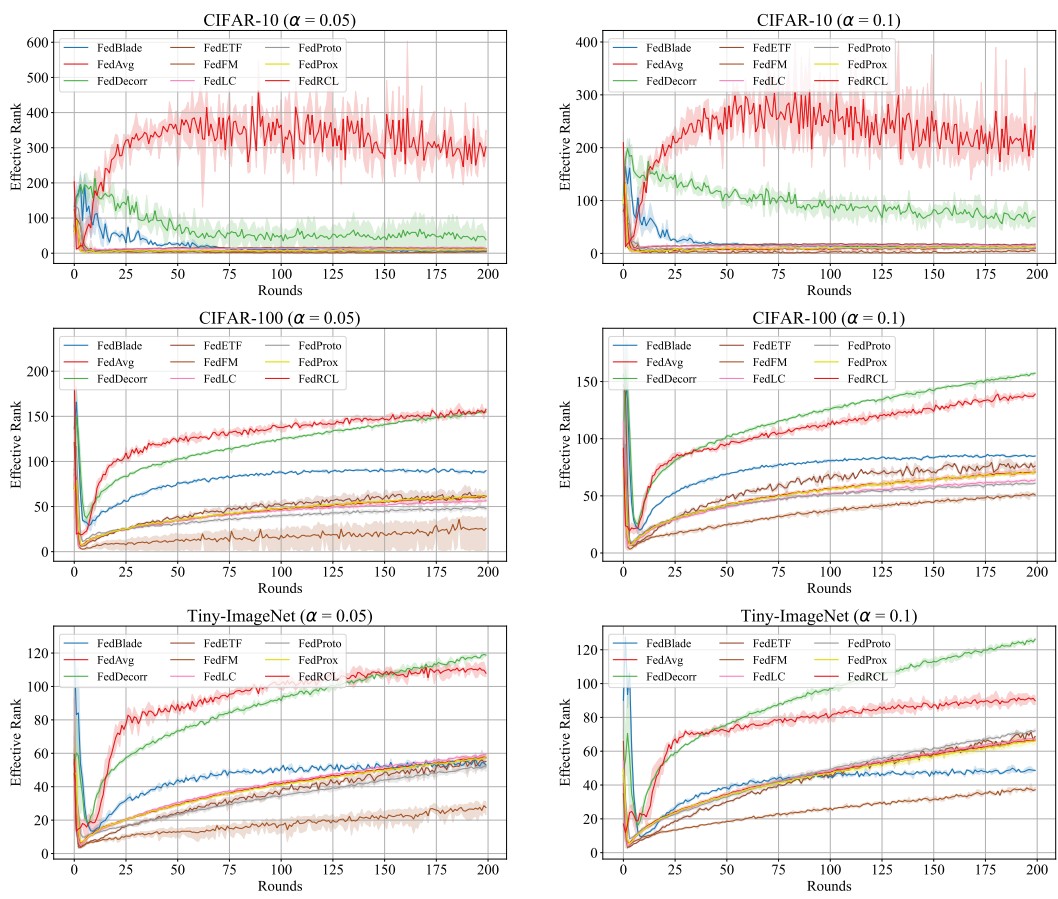

Figure 10: Effective rank under various label skew settings. The effective rank is computed in the output space of feature space.

### E.5 ABLATION STUDY ON THE NUMBER OF LOCAL EPOCHS

We conduct an ablation study on the number of local epochs $E$. We turn the local epochs $E \in \{1, 5, 10\}$ on CIFAR-10 with $\alpha$ being $\{0.05, 0.1, 0.5\}$. Tab. 9 shows that FedBlade benefits from multiple local updates (performance increases substantially from $E = 1$ to $E = 5$). However, the

gain from $E = 5$ to $E = 10$ is small relative to the additional computation cost. For this reason, we set $E = 5$ in our main experiments.

Table 9: Ablation study on the number of local epochs.

| $E$ | CIFAR-10 | | | CIFAR-100 | | | Tiny-ImageNet | | |
|---|---|---|---|---|---|---|---|---|---|
| | $Dir(0.05)$ | $Dir(0.1)$ | $Dir(0.5)$ | $Dir(0.05)$ | $Dir(0.1)$ | $Dir(0.5)$ | $Dir(0.05)$ | $Dir(0.1)$ | $Dir(0.5)$ |
| 1 | 58.33±1.45 | 64.64±1.51 | 71.33±0.48 | 31.90±0.68 | 31.73±0.53 | 33.04±0.27 | 19.03±0.32 | 18.73±0.56 | 18.25±0.41 |
| 5 | 75.83±0.70 | 81.67±0.30 | 87.90±0.14 | 54.31±0.19 | 57.92±0.15 | 62.07±0.17 | 39.43±0.17 | 41.88±0.15 | 43.63±0.25 |
| 10 | 77.90±0.82 | 82.63±0.41 | 88.65±0.22 | 54.54±0.22 | 58.30±0.20 | 61.54±0.28 | 40.98±0.18 | 43.22±0.32 | 44.40±0.31 |

### E.6 COMPUTATION COST ANALYSIS

To quantify the efficiency for LDDecorr and PBA, we report the per-iteration computation cost (ms) of each loss term. Tab. 10 shows that the computation cost of each loss component is small, indicating that FedBlade is practical in real-world scenario. Besides, different datasets measure the computation cost with different numbers of classes (i.e., {10, 100, 200}), and different architectures measure the computation cost with different representation dimensions (i.e., 512 and 1280).

Table 10: **The computation cost (ms/iteration) for each loss component**, averaged over 1000 trials. The representation dimension of MobileNetV2 is 1280, while that of ResNet-18 is 512.

| Dataset | Model | $\mathcal{L}_{sup}$ | $\mathcal{L}_{FA}$ | $\mathcal{L}_{PA}$ | $\mathcal{L}_{LDDecorr}$ |
|---|---|---|---|---|---|
| CIFAR-10 | MobileNetV2 | 0.0886 | 0.1824 | 0.1294 | 1.1522 |
| | ResNet-18 | 0.0882 | 0.1551 | 0.1255 | 0.5297 |
| CIFAR-100 | MobileNetV2 | 0.0933 | 0.2566 | 0.1373 | 1.1600 |
| | ResNet-18 | 0.0912 | 0.1606 | 0.1299 | 0.5592 |
| TinyImageNet | MobileNetV2 | 0.0884 | 0.5700 | 0.1322 | 1.1460 |
| | ResNet-18 | 0.0899 | 0.1947 | 0.1282 | 0.5245 |

### E.7 COMMUNICATION COST OF PROTOTYPES.

We compared the additional communication cost of prototypes across different datasets and architectures. For MobileNetV2, each class prototype is a 1280-dimensional float32 vector, which is only 5 KB per class; for ResNet-18, the prototype is a 512-dimensional float32 vector, requiring 2 KB per class. We report the total communication cost of all prototypes in Tab. 11. While PBA introduces an additional step for exchanging prototypes, the associated communication overhead is negligible compared to transmitting model parameters.

Table 11: The communication cost of prototypes across different datasets and architectures.

| Model | CIFAR-10 | CIFAR-100 | Tiny-ImageNet |
|---|---|---|---|
| MobileNetV2 | 50KB | 500KB | 1000KB |
| ResNet-18 | 20KB | 200KB | 400KB |

### E.8 CONVERGENCE SPEED

To quantify the convergence speed, we report the communication round at which each method first reaches the specified accuracy. For CIFAR-100, the specific accuracy values are 40% and 50%; for Tiny-ImageNet, the specific accuracy values are 20% and 30%. 200+ means the specific accuracy was not reached after 200 rounds. Benifitting from our LDDecorr and module synergy, FedBlade

achieves substantially faster convergence under various settings. In particular, we find that feature alignment methods (e.g., FedFM) converge slowly under severe label skew. This is because that, under severe label skew, dimensional collapse occurs and the prototypes used for feature alignment can be biased, which misleads the feature alignment during local training.

Table 12: Convergence speed under CIFAR-100 ($\alpha = 0.05$).

| | 40% accuracy | | 50% accuracy | |
| --- | --- | --- | --- | --- |
| | Number of rounds | Speedup | Number of rounds | Speedup |
| FedAvg | 85 | ($1.0\times$) | 184 | ($1.0\times$) |
| FedBlade | 48 | ($1.7\times$) | 105 | ($1.9\times$) |
| FedProx | 88 | ($1.0\times$) | 180 | ($1.0\times$) |
| FedLC | 68 | ($1.3\times$) | 174 | ($1.1\times$) |
| FedDecorr | 68 | ($1.3\times$) | 176 | ($1.0\times$) |
| FedRCL | 77 | ($1.1\times$) | 180 | ($1.0\times$) |
| FedProto | 85 | ($1.0\times$) | 175 | ($1.1\times$) |
| FedFM | 122 | ($1.0\times$) | 200+ | ($<0.9\times$) |
| FedETF | 74 | ($1.1\times$) | 151 | ($1.2\times$) |

Table 13: Convergence speed under CIFAR-100 ($\alpha = 0.1$).

| | 40% accuracy | | 50% accuracy | |
| --- | --- | --- | --- | --- |
| | Number of rounds | Speedup | Number of rounds | Speedup |
| FedAvg | 60 | ($1.0\times$) | 117 | ($1.0\times$) |
| FedBlade | 40 | ($1.5\times$) | 78 | ($1.5\times$) |
| FedProx | 60 | ($1.0\times$) | 117 | ($1.0\times$) |
| FedLC | 56 | ($1.1\times$) | 110 | ($1.1\times$) |
| FedDecorr | 53 | ($1.1\times$) | 111 | ($1.1\times$) |
| FedRCL | 53 | ($1.1\times$) | 101 | ($1.2\times$) |
| FedProto | 60 | ($1.0\times$) | 111 | ($1.1\times$) |
| FedFM | 71 | ($0.8\times$) | 131 | ($0.9\times$) |
| FedETF | 57 | ($1.1\times$) | 111 | ($1.1\times$) |

Table 14: Convergence speed under CIFAR-100 ($\alpha = 0.5$).

| | 40% accuracy | | | 50% accuracy | | |
|---|---|---|---|---|---|---|
| | Number of rounds | | Speedup | Number of rounds | | Speedup |
| FedAvg | 43 | | ($1.0\times$) | 76 | | ($1.0\times$) |
| FedBlade | 34 | | ($1.3\times$) | 58 | | ($1.3\times$) |
| FedProx | 43 | | ($1.0\times$) | 77 | | ($1.0\times$) |
| FedLC | 41 | | ($1.0\times$) | 75 | | ($1.0\times$) |
| FedDecorr | 40 | | ($1.1\times$) | 79 | | ($1.0\times$) |
| FedRCL | 40 | | ($1.1\times$) | 70 | | ($1.1\times$) |
| FedProto | 41 | | ($1.0\times$) | 71 | | ($1.1\times$) |
| FedFM | 41 | | ($1.0\times$) | 70 | | ($1.1\times$) |
| FedETF | 53 | | ($0.8\times$) | 83 | | ($0.9\times$) |

Table 15: Convergence speed under Tiny-ImageNet ($\alpha = 0.05$).

| | 20% accuracy | | | 30% accuracy | | |
|---|---|---|---|---|---|---|
| | Number of rounds | | Speedup | Number of rounds | | Speedup |
| FedAvg | 70 | | ($1.0\times$) | 141 | | ($1.0\times$) |
| FedBlade | 41 | | ($1.7\times$) | 81 | | ($1.7\times$) |
| FedProx | 70 | | ($1.0\times$) | 150 | | ($0.9\times$) |
| FedLC | 49 | | ($1.4\times$) | 102 | | ($1.4\times$) |
| FedDecorr | 45 | | ($1.6\times$) | 101 | | ($1.4\times$) |
| FedRCL | 88 | | ($0.8\times$) | 166 | | ($0.8\times$) |
| FedProto | 82 | | ($0.9\times$) | 170 | | ($0.9\times$) |
| FedFM | 130 | | ($0.5\times$) | 200+ | | ($<0.7\times$) |
| FedETF | 60 | | ($1.2\times$) | 116 | | ($1.2\times$) |

Table 16: Convergence speed under Tiny-ImageNet ($\alpha = 0.1$).

| | 20% accuracy | | | 30% accuracy | | |
|---|---|---|---|---|---|---|
| | Number of rounds | | Speedup | Number of rounds | | Speedup |
| FedAvg | 46 | | ($1.0\times$) | 104 | | ($1.0\times$) |
| FedBlade | 37 | | ($1.2\times$) | 67 | | ($1.5\times$) |
| FedProx | 51 | | ($0.9\times$) | 104 | | ($1.0\times$) |
| FedLC | 40 | | ($1.1\times$) | 81 | | ($1.3\times$) |
| FedDecorr | 37 | | ($1.2\times$) | 79 | | ($1.3\times$) |
| FedRCL | 63 | | ($0.7\times$) | 121 | | ($0.9\times$) |
| FedProto | 57 | | ($0.8\times$) | 114 | | ($0.9\times$) |
| FedFM | 74 | | ($0.6\times$) | 157 | | ($0.7\times$) |
| FedETF | 50 | | ($0.9\times$) | 92 | | ($0.9\times$) |

Table 17: Convergence speed under Tiny-ImageNet ($\alpha = 0.5$).

| | 20% accuracy | | 30% accuracy | |
| --- | --- | --- | --- | --- |
| | Number of rounds | Speedup | Number of rounds | Speedup |
| FedAvg | 37 | ( $1.0\times$ ) | 75 | ( $1.0\times$ ) |
| FedBlade | 39 | ( $0.9\times$ ) | 66 | ( $1.1\times$ ) |
| FedProx | 37 | ( $1.0\times$ ) | 77 | ( $1.0\times$ ) |
| FedLC | 35 | ( $1.1\times$ ) | 71 | ( $1.1\times$ ) |
| FedDecorr | 30 | ( $1.2\times$ ) | 60 | ( $1.25\times$ ) |
| FedRCL | 46 | ( $0.8\times$ ) | 82 | ( $0.9\times$ ) |
| FedProto | 39 | ( $0.9\times$ ) | 76 | ( $1.0\times$ ) |
| FedFM | 43 | ( $0.9\times$ ) | 89 | ( $0.8\times$ ) |
| FedETF | 46 | ( $0.8\times$ ) | 79 | ( $0.9\times$ ) |

# F  THE USE OF LARGE LANGUAGE MODELS (LLMS)

We used LLMs **only for language polishing**. All contents were line-by-line verified, including contents generated by LLMs.

