# OpenReview forum: "Stabilizing Heterogeneous Federated Learning via Feature Decorrelation and Bidirectional Alignment"
_ICLR.cc/2026/Conference — Submitted to ICLR 2026_

### Official Review · Reviewer_BH9Y · 2025-10-20

**Soundness:** 3
**Presentation:** 2
**Contribution:** 2
**Rating:** 4
**Confidence:** 4

**Summary:**

- Connects the the non-IID problem in Federated Learning (FL) to the collapse of singular values. While the connection of the singular values and the collapse that happens in non-IID FL is well documented in prior work, the authors make a point that previous studies do not focus on preventing collapse of the smaller singular values.

- Introduces a new decorrelation method of minimizing the negative log-determinant of the representation correlation matrix. They argue that this approach prevents the collapse of the of the smaller singular values, which expands the useable feature space. With the expanded space, they argue that their alignment method improves performance.

**Strengths:**

- The authors do well in emphasizing the need to focus on the small singular values.
- The design of the decorrelation loss is quite clever. I don't think I've seen the determinant being used for a decorrelation loss, but this makes sense when we're focusing on the smaller singular values.
- Decent results in terms of performance and speedup.

**Weaknesses:**

- The main problem I have with this paper is the lack of transparency for the efficacy of the method. The authors' central claim is that their method increases effective rank, as their loss more harshly penalizes the small singular value. Since the authors claim this, they MUST also provide the effective rank for other methods, so readers can truly see that their method is effective not only in increasing the effective rank, but also the performance (which they do show). From Figure 4, they only show ablations of their own method. I could not find the effective rank for other methods.
- I also think the authors have missed two key citations [1], [2]. [1] introduced the idea of keeping the classifier fixed, because the classifier leads to much bias. [2] made connections to the singular values, debiasing, and increasing feature space.
- The core idea of this paper is also quite similar to that of [2]. FedUV uses Uniformity to increase the feature space, and Variance to debias the classifier. I feel this paper needs to frame their paper in a different light to highlight their novelty (likely the decorrelation loss).

- I feel the name of 'bidirectional alignment' is quite poor. It's alignment to the projector and the output of the encoder. There's no 'bi-directional' anything going on. Maybe 'bi-focused'? I'm sure there would be a better name. The goal is the convey that you are targeting two places in the network, not two directions.

[1] Fedbabu: Towards enhanced representation for federated image classification, ICLR 2022.
[2] FedUV: Uniformity and Variance for Heterogeneous Federated Learning, CVPR 2024

**Questions:**

1. Can the authors provide more transparency into the effective rank for other methods? They should also provide their rationale backed by as much data as they can provide regarding this point.
2. Can the authors frame their method in a more novel light, by first citing the very relevant papers mentioned above?

---

> ### Author Response · Authors · 2025-11-20
> **Response to Reviewer BH9Y 1/2**
>
> Dear Reviewer BH9Y,
>
> Thanks for your valuable comments and kind words to our work. Below we address specific questions:
>
> > W1 & Q1: The main problem I have with this paper is the lack of transparency for the efficacy of the method. The authors' central claim is that their method increases effective rank, as their loss more harshly penalizes the small singular value. Since the authors claim this, they MUST also provide the effective rank for other methods, so readers can truly see that their method is effective not only in increasing the effective rank, but also the performance (which they do show). From Figure 4, they only show ablations of their own method. I could not find the effective rank for other methods.
>
> **A:** Thank you for pointing out the need for clearer comparisons of effective rank. In Figure 4, we evaluated effective rank in **the output space of projector rathen than that of feature extractor**. This can measure the final embedding space used by the classifier. Comparing effective rank across baselines without a projector would mix architectural effects with the impact of decorrelation, making the comparison uninterpretable. For this reason, **Figure 4 focuses exclusively on decorrelation losses under the same architecture to isolate their effect**.
>
> To provide broader transparency, we have added effective-rank results for all methods in the revised manuscript (Appendix E.4), **computed in the output space of feature extractor, which is shared across architectures**. These results show that FedBlade increases effective rank relative to some baselines, but these results also indicate that **effective rank and accuracy are not strictly monotonic**. Once feature diversity is sufficient, **excessive rank expansion can degrade class structure**. This phenomenon can be supported by the observations in [1], where over-whitened features can break the potential manifold the examples in the same class belong to. Besides, appropriate effective rank can be also supported by neural collapse [2], where **good generalization is associated with structured high-dimensional geometries** (i.e., simplex ETF), rather than arbitrarily increasing the dimensionality of representations.
>
> Importantly, combining LDDecorr with PBA yields **both higher effective rank and a more structured ETF-like geometry**, explaining why FedBlade achieves stronger performance. We have clarified these points and included the full effective-rank comparisons in the revised manuscript.
>
> > W2: I also think the authors have missed two key citations [1], [2]. [1] introduced the idea of keeping the classifier fixed, because the classifier leads to much bias. [2] made connections to the singular values, debiasing, and increasing feature space.
>
> **A:** Thank you for pointing out these relevant works. We have added these citations and clarified their relation to our approach in the revised manuscript.
>
> Although the classifier is fixed both in FedBlade and FedBABU, they target different aspects of federated learning. FedBlade imposes a structured equiangular geometry that aligns class directions across clients and stabilizes representation learning under label skew, while FedBABU focuses on decoupling body and head optimization for local adaptation without controlling the representation geometry. As a result, **FedBlade is better suited for settings with severe label skew and dimensional collapse**.
>
> Regarding FedUV, **although both FedUV and FedBlade consider spectral properties, they operate on different objects and address different problems**. FedUV analyzes the singular values of the **classifier weight matrix** and regularizes logits to prevent classifier degeneration. In contrast, FedBlade focuses on the **representation correlation matrix**, where severe label skew induces dimensional collapse. LDDecorr directly reshapes this correlation spectrum via log-determinant regularization, boosting small singular values and expanding effective rank. This is an effect not addressed by the pairwise uniformity regularizer of FedUV. Furthermore, FedUV does not impose discriminative geometric structure on classifier directions, whereas FedBlade leverages bilateral alignment and a fixed ETF classifier to improve consistency across clients.
>
> **Reference:**
>
> [1] Weng, X., Huang, L., Zhao, L., Anwer, R., Khan, S. H., & Shahbaz Khan, F. (2022). An investigation into whitening loss for self-supervised learning. Advances in Neural Information Processing Systems, 35, 29748-29760.
>
> [2] Papyan, V., Han, X. Y., & Donoho, D. L. (2020). Prevalence of neural collapse during the terminal phase of deep learning training. Proceedings of the National Academy of Sciences, 117(40), 24652-24663.

---

> ### Author Response · Authors · 2025-11-20
> **Response to Reviewer BH9Y 2/2**
>
> > W3: The core idea of this paper is also quite similar to that of [2]. FedUV uses Uniformity to increase the feature space, and Variance to debias the classifier. I feel this paper needs to frame their paper in a different light to highlight their novelty (likely the decorrelation loss).
>
> **A:** Thank you for your constructive comment. We would like to clarify that **FedUV and our FedBlade analyze different spectral objects and address different problems**.
>
> FedUV applies Uniformity and Variance regularizers on pairwise features and logits, aiming to enlarge prediction variance and prevent classifier degeneration. However, it does not operate on **the spectral structure of the representation space**. In contrast, the central innovation of FedBlade lies in LDDecorr, which directly targets the singular values of the representation correlation matrix. This log-determinant decorrelation loss boosts small singular values, explicitly increases effective rank, and mitigates dimensional collapse. These are capabilities that FedUV do not provide. Moreover, FedBlade incorporates an ETF classifier and prototype-based alignment to impose **a globally consistent discriminative geometry across clients**, which is also absent in FedUV.
>
> We have revised the manuscript to more clearly highlight our novelty.
>
> > W4: I feel the name of 'bidirectional alignment' is quite poor. It's alignment to the projector and the output of the encoder. There's no 'bi-directional' anything going on. Maybe 'bi-focused'? I'm sure there would be a better name. The goal is the convey that you are targeting two places in the network, not two directions.
>
> **A:** Thank you for the suggestion regarding naming. To avoid the confusion introduced by "bidirectional", we adopt the term **bilateral alignment**, which more accurately reflects the role of PBA in our framework. Specifically, the global prototype acts as a bridge between two components of the representation pipeline, i.e., the feature extractor and the projector, ensuring that both are aligned with a consistent geometric target. The term "bilateral" emphasizes this alignment across these two modules without implying directional flows. The manuscript has been updated accordingly.
>
> > Q2: Can the authors frame their method in a more novel light, by first citing the very relevant papers mentioned above?
>
> **A:** Thank you for the helpful suggestion. In the revised manuscript, we have incorporated the relevant prior works and clarified their relationship to our approach. We highlight that, unlike methods that freeze the classifier or apply pairwise uniformity/variance regularizers, FedBlade operates directly on the spectral structure of the representation correlation matrix through LDDecorr and introduces a prototype-guided alignment mechanism that enforces a globally consistent ETF-based geometry. These clarifications more clearly frame the novelty of our method while situating it within the broader literature.

---

> > ### Comment · Reviewer_BH9Y · 2025-11-24
> >
> > Thank you authors for the response and the updated paper. I appreciate the new analysis. I also agree with their interpretations and it is intuitive that having a higher rank is not always good.
> >
> > However, I am concerned that these results were not shown in the original paper, and that these arguments for the over-rank are only shown in the new version. Not only that, these results are still not well argued in the main text.
> >
> > The main text leads the reader to believe that higher rank is always beneficial, yet the nuance of over-rank is missed when only Figure 1 and Figure 2 is shown.
> >
> > Therefore, I will keep my original score of 4. The reason is that I believe a change to address this would require the restructuring of the entire paper.

---

> > > ### Author Response · Authors · 2025-11-25
> > > **Response to Reviewer BH9Y**
> > >
> > > Thanks for your reply and helpful feedback. Following your suggestion, we have revised our manuscript. In the revised manuscript, we explicitly clarify in the main text that effective rank and accuracy are not strictly monotonic, and that excessively large rank can distort intra-class structure. To support this discussion, we now include the effective rank of all methods on CIFAR-100 ($\alpha$=0.05) and Tiny-ImageNet ($\alpha$=0.05) directly in the main paper, with full results for all datasets provided in Appendix E.4. These additions clarify that FedBlade improves effective rank while avoiding unnecessary over-expansion once sufficient feature diversity has been achieved.
> > >
> > > We hope these revisions address the concern regarding how effective rank relates to accuracy in the main text.

---

### Official Review · Reviewer_CBFg · 2025-10-30

**Soundness:** 4
**Presentation:** 4
**Contribution:** 3
**Rating:** 6
**Confidence:** 4

**Summary:**

This paper proposes FedBlade, a federated learning framework addressing label skew via two main components: (1) LDDecorr, a log-determinant based feature decorrelation regularizer that produces exponential gradients to impose stronger penalties on small singular values, thereby mitigating dimensional collapse better than FedDecorr; and (2) PBA, a prototype-guided bidirectional alignment mechanism that aligns the feature extractor and projector with a shared ETF classifier through global prototypes. The whole method builds upon FedETF and FedDecorr, aiming to enforce neural collapse and reduce dimensional collapse, under label skew FL.

**Strengths:**

1. The motivation and intuition of this work are clear and sound, which provides a well-articulated motivation for addressing dimensional collapse and classifier bias in federated learning.

2. This work has solid theoretical grounding. Theoretical analysis connecting the log-determinant term to spectrum isotropy and effective rank is insightful.

3. Using global prototypes as bridges between modules (extractor–projector–ETF classifier) is a natural and elegant idea that improves interpretability.

4. The paper is well-structured, and the presentation is clear and easy to follow.

**Weaknesses:**

1. The proposed method is an incremental improvement over FedETF and FedDecorr, mainly combining a modified decorrelation loss and a prototype-based alignment. The novelty is moderate.

2. The improvement is limited and often small, especially under full client participation.

3. The ablation results are somewhat puzzling: on CIFAR-100, LDDecorr improves accuracy while PBA reduces it; on Tiny-ImageNet, the opposite trend occurs. This suggests dataset-specific sensitivity or complex interactions between the two modules that are not well explained in the paper.
Interestingly, when both components are applied together, the overall performance improves significantly and consistently across datasets, implying that LDDecorr and PBA may complement each other in a deeper way. For instance, LDDecorr might enhance the representation isotropy that PBA relies on for effective alignment, while PBA may in turn stabilize the projection space required for LDDecorr to operate effectively. A more detailed and theoretical analysis of this potential synergy would strengthen the paper’s understanding of why the two modules jointly work better than each alone.

4. Although LDDecorr theoretically reduces computational cost via Cholesky decomposition, there exists a bad case:  on Tiny-ImageNet with α=0.5, where FedDecorr converges faster. Authors should explain why.


5. The experimental scope is narrow. Experiments use only MobileNetV2 and three vision benchmarks. Testing on additional architectures or non-vision datasets would strengthen the generality claim.

6. Lack of cost metrics. No wall-clock, FLOPs, or communication-overhead analysis is provided to substantiate claims of improved efficiency.

**Questions:**

1. The improvements under full client participation seem limited. Could the authors provide a detailed analysis of why FedBlade’s advantages diminish in this setting?

2. The ablation trends are inconsistent across datasets (Table 3). Can the authors explain why LDDecorr and PBA sometimes have opposite effects? Is this due to prototype drift, sensitivity to γ/β, or data distribution differences?

3. Could the authors show convergence curves or wall-clock time comparisons for α=0.5 to better support the claim of “faster convergence”?

4. How sensitive are the results to hyperparameters β, γ, and τ?

5. Does PBA introduce additional communication overhead when aggregating global prototypes, and if so, how significant is it?

6. The paper mentions that FedBlade “enforces neural collapse.” Could the authors provide quantitative evidence (e.g., NC1–NC4 metrics) to substantiate this claim?

---

> ### Author Response · Authors · 2025-11-20
> **Response to Reviewer CBFg 1/3**
>
> Dear Reviewer CBFg,
>
> Thanks for your constructive comments and kind words to our work. Below we address specific questions:
>
> > W1: The proposed method is an incremental improvement over FedETF and FedDecorr, mainly combining a modified decorrelation loss and a prototype-based alignment. The novelty is moderate.
>
> **A:** Thank you for the comment. We believe there may be some misunderstandings. Our contributions go beyond a simple combination, and we clarify the novelty below.
>
> **Our LDDecorr introduces a fundamentally different collapse mitigation mechanism.** FedDecorr applies a Frobenius penalty that treats all singular values uniformly and provides only a **linear penalty** on small singular values, which limits its ability to recover the ambient feature space. LDDecorr maximizes the log-determinant of the feature correlation matrix, yielding **exponential gradients** that impose extremely strong penalties on small singular values. This effectively prevents dimensional collapse, **a limitation unaddressed in prior FL decorrelation methods**. As noted by Reviewer BH9Y: "The design of the decorrelation loss is quite clever. I don't think I've seen the determinant being used for a decorrelation loss, but this makes sense when we're focusing on the smaller singular values."
>
> **PBA introduces synergy-aware alignment across the feature extractor, projector, and ETF classifier.** FedETF fixes the classifier but does not ensure consistency between feature extractor, projector, and classifier. Existing prototype-based methods also do not enforce such joint alignment. PBA uses global prototypes as a geometric bridge to align all modules simultaneously, promoting consistency across clients.
>
> **The two components are complementary rather than additive.** As analyzed in the Introduction, LDDecorr and PBA address *distinct yet interdependent challenges* under label skew. LDDecorr preserves expressive capacity by preventing rank collapse, ensuring prototypes remain informative. PBA, in turn, provides structured ETF geometry that counteracts potential distortions introduced by strong decorrelation. Our ablations confirm that their combination yields the largest gain, indicating **genuine synergy rather than incremental stacking**.
>
> In summary, FedBlade introduces (1) a new feature decorrelation method tailored for collapse mitigation in FL, (2) a synergy-aware prototype alignment method, and (3) an integrated design that achieves substantially stronger robustness under label skew. We have revised the manuscript to highlight these contributions more clearly.
>
> > W2 & Q1: The improvement is limited and often small, especially under full client participation.
>
> **A:** Thank you for the observation. We clarify this from three perspectives.
>
> (1) Level of heterogeneity. FedBlade is designed to address challenges that arise primarily under **severe label-skew**, i.e., dimensional collapse and inconsistent feature extractors and classifiers. When heterogeneity is mild, as in full participation or $Dir(0.5)$, these issues are reduced, so the improvement is limited.
>
> (2) Number of classes. Prototype-based alignment is more impactful when the number of classes $C$ is large. With an ETF classifier, **decision boundaries become geometrically narrower as $C$ increases**, making classification more sensitive to feature bias. Thus, PBA brings larger improvements on datasets like CIFAR-100 and Tiny-ImageNet than on CIFAR-10.
>
> (3) Client participation. **Full participation provides each client with more local data and consistent updates across rounds, leading to more stable optimization and less prototype shift.** As a result, the relative benefit of FedBlade diminishes. In contrast, partial participation, which is common in real-world FL due to computation and communication constraints, creates stronger skew across clients. In this scenario, FedBlade shows clear advantages.
>
> We have added clarification in the manuscript to better contextualize these results.

---

> ### Author Response · Authors · 2025-11-20
> **Response to Reviewer CBFg 2/3**
>
> > W3 & Q2: The ablation results are somewhat puzzling: .... why the two modules jointly work better than each alone.
>
> **A:** Thanks for your insightful comment. We agree that the ablation results reflect complex interactions between LDDecorr and PBA, and we have expanded our analysis in the revised manuscript. The two modules address distinct yet  interdependent challenges under label skew.
>
> LDDecorr mitigates dimensional collapse by expanding the effective rank of the representation space. This preserves discriminative directions that are essential for generating reliable global prototypes. However, strong decorrelation alone does not guarantee a well-structured feature space, and **without additional constraints the resulting feature space can remain poorly aligned with the ETF classifier**.
>
> PBA promotes neural-collapse–like structure by aligning feature extractor, projector, and the ETF classifier. This improves the consistency across clients but relies on sufficiently informative features. **When features have collapsed, prototype quality deteriorates and alignment becomes unreliable**.
>
> **The synergy emerges because each module provides what the other lacks.** LDDecorr ensures that the representation space retains enough dimensionality for PBA to generate meaningful prototypes, while PBA imposes structured geometry that counteracts the potential instability caused by strong decorrelation. This explains why the combined method consistently performs best across datasets.
>
> We then analyze the ablation results. On datasets with many classes (e.g., Tiny-ImageNet), dimensional collapse is more harmful because decision boundaries are narrower and prototype generation is more sensitive to the reduced dimensionality. Here LDDecorr contributes more. On datasets with fewer classes (e.g., CIFAR-100), LDDecorr may make it difficult to form a well-structed embedding space that is required for the ETF classifier. Thus, PBA is crucial.
>
> When used together, the two components correct the weaknesses of each other. We sincerely appreciate your comment, which helps us better articulate and highlight the deeper synergy. We have incorporated these explanations into the revised manuscript.
>
> > W4: Although LDDecorr theoretically reduces computational cost via Cholesky decomposition, there exists a bad case: on Tiny-ImageNet with α=0.5, where FedDecorr converges faster. Authors should explain why.
>
> **A:** Thank you for pointing it out. In Tiny-ImageNet with $\alpha$ = 0.5, FedDecorr shows faster convergence during the early training rounds, as seen in Fig. 8. This behavior is not related to the computational complexity of LDDecorr, but is instead driven by early-stage optimization dynamics. Prototype-based alignment methods typically spend the early phase of training **establishing reliable global prototypes**, and this phase becomes longer when the number of classes is large. Once well-separated prototypes are formed, the alignment becomes significantly more effective, so **FedBlade accelerates and surpasses FedDecorr in later rounds**.
>
> > W5: The experimental scope is narrow... Testing on additional architectures or non-vision datasets would strengthen the generality claim.
>
> **A:** Thanks for your constructive comment. Our work focuses on dimensional collapse and classifier bias, which are especially prevalent in vision models using deep feature extractor. Thus, we evaluate FedBlade primarily on standard vision benchmarks. To strengthen the architectural diversity of our experiments, **we additionally include results on ResNet-18**, which differs substantially from MobileNetV2 in structure and representation dimensionality. As shown in the table, FedBlade consistently improves performance.
>
> |Method|CIFAR10(0.05)|CIFAR10(0.1)|CIFAR10(0.5)|CIFAR100(0.05)|CIFAR100(0.1)|CIFAR100(0.5)|TinyImageNet(0.05)|TinyImageNet(0.1)|TinyImageNet(0.5)|
> |---|---|---|---|---|---|---|---|---|---|
> |FedAvg|62.09±3.67|73.78±4.11|88.69±0.57|57.08±0.47|60.81±0.44|63.44±0.25|39.70±0.48|43.29±0.25|46.29±0.20|
> |FedProx|62.76±3.97|74.68±3.51|88.65±0.55|56.97±0.33|60.79±0.38|63.19±0.25|39.84±0.40|43.45±0.68|46.38±0.32|
> |FedLC|79.83±0.62|84.68±0.31|89.20±0.17|56.62±0.64|58.77±0.27|61.93±0.20|42.65±0.17|45.66±0.17|47.37±0.21|
> |FedDecorr|67.05±3.33|76.53±3.91|88.74±0.41|54.72±0.38|57.40±0.35|58.14±0.32|40.17±0.46|42.92±0.27|44.83±0.19|
> |FedRCL|63.83±3.02|71.07±1.81|86.84±0.34|52.42±0.31|58.70±0.38|63.49±0.26|37.00±0.33|41.61±0.40|45.53±0.33|
> |FedProto|61.58±3.03|74.41±4.08|89.16±0.47|56.18±0.71|59.69±0.43|64.27±0.25|39.43±0.72|43.53±0.33|47.82±0.24|
> |FedFM|63.79±2.83|75.48±4.73|88.88±0.85|55.43±0.56|61.42±0.68|**67.40±0.24**|34.12±1.25|41.54±0.71|**48.08±0.29**|
> |FedETF|80.41±0.61|85.08±0.38|89.36±0.13|57.61±0.70|60.09±1.21|60.92±1.23|42.79±0.56|45.41±0.18|47.08±0.52|
> |FedBlade|**81.41±0.50**|**85.66±0.13**|**90.25±0.13**|**60.56±0.20**|**63.94±0.27**|65.55±0.13|**43.43±0.28**|**45.84±0.26**|47.94±0.25|

---

> ### Author Response · Authors · 2025-11-20
> **Response to Reviewer CBFg 3/3**
>
> > W6: Lack of cost metrics. No wall-clock, FLOPs, or communication-overhead analysis is provided to substantiate claims of improved efficiency.
>
> **A:** Thanks for your valuable comment. We provide several forms of cost analysis in the paper. First, Table 2, Appendix E.3 and Appendix E.8 show that **FedBlade requires substantially fewer communication rounds** to reach the same accuracy as competing methods, directly reducing total communication and local training cost. **Because local training and communication dominate the FL training time, this reduction provides a practical efficiency benefit.**
>
> Second, **the overhead introduced by our loss terms is small**. The table below reports the per-iteration computation time (ms) of each component. These costs are minor relative to the communication cost per round.
>
> |Dataset|Model|$\mathcal{L}_{sup}$|$\mathcal{L}_{FA}$|$\mathcal{L}_{PA}$|$\mathcal{L}_{LDDecorr}$|
> |---|---|---|---|---|---|
> |CIFAR-10|MobileNetV2|0.0886|0.1824|0.1294|1.1522|
> |CIFAR-10|ResNet-18|0.0882|0.1551|0.1255|0.5297|
> |CIFAR-100|MobileNetV2|0.0933|0.2566|0.1373|1.1600|
> |CIFAR-100|ResNet-18|0.0912|0.1606|0.1299|0.5592|
> |TinyImageNet|MobileNetV2|0.0884|0.5700|0.1322|1.1460|
> |TinyImageNet|ResNet-18|0.0899|0.1947|0.1282|0.5245|
>
> > Q3: Could the authors show convergence curves or wall-clock time comparisons for $\alpha$=0.5 to better support the claim of “faster convergence”?
>
> **A:** Thanks for your comment. We have included convergence curves for $\alpha = 0.5$ in Appendix E.3. These curves show that FedBlade achieves faster convergence on three heterogeneity settings. The only exception is the early stage on Tiny-ImageNet ($\alpha$ = 0.5), where FedDecorr increases more quickly, but FedBlade overtakes it and achieves higher final performance. More details are provided in our response to W4.
>
> > Q4: How sensitive are the results to hyperparameters $\beta$, $\gamma$, and $\tau$?
>
> **A:** The sensitivity analyses of $\beta$, $\gamma$ and $\tau$ are provided in Appendix D. Across all three hyperparameters, the results show that FedBlade remains stable within a reasonable range.
>
> > Q5: Does PBA introduce additional communication overhead when aggregating global prototypes, and if so, how significant is it?
>
> **A:** PBA introduces an additional communication cost for exchanging class prototypes, but the overhead is small. With $C$ classes and a $d$-dimensional float32 prototype, the per-round communication cost is $4Cd$ bytes. In our setup with MobileNetV2 ($d$ = 1280), this corresponds to approximately 50 KB (CIFAR-10), 500 KB (CIFAR-100), and 1000 KB (Tiny-ImageNet) per round. This cost is much smaller than the communication cost for transmitting model parameters. Thus, **the communication added by PBA is negligible in practical FL settings**. We analyze the communication cost of PBA across different datasets and model architectures.
>
> |Model|CIFAR-10|CIFAR-100|Tiny-ImageNet|
> |---|---|---|---|
> |MobileNetV2|50KB|500KB|1000KB|
> |ResNet-18|20KB|200KB|400KB|
>
> > Q6: The paper mentions that FedBlade “enforces neural collapse.” Could the authors provide quantitative evidence (e.g., NC1–NC4 metrics) to substantiate this claim?
>
> **A:** Thanks for you constructive comment. To demonstrate that PBA encourages neural collapse, we quantify two standard metrics on CIFAR-100: within-class variance (NC1) and deviation from the simplex ETF structure (NC2), where lower values indicate stronger neural-collapse behavior. As shown in the below table, **adding PBA consistently reduces both NC1 and NC2** across all heterogeneity settings, demonstrating that PBA promotes **tighter class clusters and more ETF-like feature geometry**. These results provide direct quantitative evidence that **PBA contributes to the formation of neural collapse**.
>
> |Dir($\alpha$)|Method|NC1 $\downarrow$|NC2 $\downarrow$|
> |---|---|---|---|
> |Dir(0.05)|w/o PBA|0.6425|22.3306|
> |Dir(0.05)|w/ PBA |0.5669|18.5304|
> |Dir(0.1) |w/o PBA|0.6367|21.4161|
> |Dir(0.1) |w/ PBA |0.5268|17.0571|
> |Dir(0.5) |w/o PBA|0.6260|20.8275|
> |Dir(0.5) |w/ PBA |0.4762|16.3720|

---

### Official Review · Reviewer_1H5i · 2025-11-03

**Soundness:** 1
**Presentation:** 3
**Contribution:** 2
**Rating:** 2
**Confidence:** 5

**Summary:**

This paper proposes a two-pronged approach to dealing with label skew in Federated Learning for natural image classification. The first component modifies the regularization term introduced in FedDecorr (which encourages different dimensions of representations to be uncorrelated) from the Frobenius norm of the correlation matrix to the negative log-determinant of the same. The second component consists of two sub-components: (i) the projector & frozen ETF classifier head first proposed in FedETF that leverages the neural collapse phenomenon to reduce classifier bias (ii) global class prototypes (like in FedProto) that both the projector and main network are aligned with via separate alignment losses. Experiments are conducted on CIFAR10/CIFAR100 and TinyImageNet, under Dirichlet non-IIDness with alpha values 0.05, 0.1, 0.5, using an untrained MobileNetV2, against 8 baseline methods. There is also an ablation experiment of the components, and sensitivity analysis of the introduced hyper-parameters.

**Strengths:**

The paper benefits from a clear motivation, namely mitigating dimensional collapse and addressing classifier bias at the same time. Figure 1 helps build intuition to this effect. The idea itself is interesting even if its not that novel (see W1). The experimental setup for the most part falls in line with similar papers, and the range of compared baselines is pretty diverse. The paper is generally well-written, with good pacing, and the target problem is an important one.

**Weaknesses:**

I appreciate the authors' efforts in tackling the important problem of heterogeneity in federated learning. However, I have some concerns that I hope can be addressed to strengthen the contribution:

1. Theoretical novelty and experimental depth: The paper presents an interesting combination of previous work, but I found myself wanting more depth in certain areas:
* The approach builds upon FedDecorr (replacing the Frobenius norm with log-determinant) and FedETF (incorporating prototypes similar to FedProto/FedNH [3]). Given this foundation, I believe the paper would benefit from either theoretical justification for why this combination is effective or more comprehensive empirical validation.
* The log-determinant technique for matrix rank minimization is well-established [1], so it would be helpful to clarify the specific novelty here.
* While the methodology section presents various formulas, some appear to go unused in the final approach. Additionally, formal analysis (e.g., convergence guarantees or theoretical properties) would strengthen the contribution.
* The experimental scope could be broadened to include more diverse heterogeneity scenarios (e.g., domain shift, real-world datasets, less extreme α settings). I also noticed the model backbone doesn't leverage recent FL research showing that pre-trained models and architectures without BatchNorm can significantly mitigate non-IID performance issues.

2. Related work positioning: The paper would benefit from a broader contextualization within the federated optimization literature:
* While the focus on dimensional collapse and classifier bias is valuable, the heterogeneity challenge in FL has been extensively studied from multiple angles. I'd suggest considering [2] as well, which shares conceptual similarities.
* Recent work has demonstrated that BatchNorm usage and non-pretrained models contribute significantly to performance degradation under non-IID data. Acknowledging this literature and clarifying when/where the proposed approach is most advantageous would help readers understand the method's positioning. Happy to provide some starter references on this if needed.

3. Experimental design considerations: I have several concerns about the experimental setup that might affect the interpretation of results:

* The choice of E=5 local epochs for 100 rounds is interesting. Since higher E under non-IID conditions often degrades performance, could you provide an ablation study (similar to FedDecorr's analysis) or justification for this choice versus E=1? I also noticed in Figures 3, 4, and 7 that algorithms haven't converged at T=100. Would a lower E with longer training change the conclusions and result ranking?

* Under full participation (Table 6), improvement is marginal, with baselines outperforming or performing within the variance range of the proposed method, despite being simpler. Could you discuss these results?
* I noticed FedDecorr's $\beta$ is set to 10 here, while their paper used 0.1 under similar settings. Could you clarify this 100x difference to ensure fair comparison?
* Regarding Figures 5 and 6: the 10% Y-axis increments make the sensitivity appear much lower than it really is, can the authors provide a rendering where the increments are more in line with the performance difference between the proposed method and baselines (e.g. 2% increments)? This would help contextualize the sensitivity better. If test set tuning was used for these hyperparameters, as it appears to be the case, this might disadvantage simpler baselines with fewer hyperparameters. Can the authors comment on this?

## Minor corrections
- Typo line 108 Relate Work -> Related Work
- Typo line 237 definded -> defined

I'm happy to discuss these points further and would be open to reconsidering my assessment if these concerns can be addressed.

## References

[1] Fazel, M., Hindi, H. and Boyd, S.P., 2003, June. Log-det heuristic for matrix rank minimization with applications to Hankel and Euclidean distance matrices. In Proceedings of the 2003 American Control Conference, 2003. (Vol. 3, pp. 2156-2162). IEEE.

[2] Guo, Y., Tang, X. and Lin, T., 2023, July. Fedbr: Improving federated learning on heterogeneous data via local learning bias reduction. In International conference on machine learning (pp. 12034-12054). PMLR.

[3] Dai, Y., Chen, Z., Li, J., Heinecke, S., Sun, L. and Xu, R., 2023, June. Tackling data heterogeneity in federated learning with class prototypes. In Proceedings of the AAAI Conference on Artificial Intelligence (Vol. 37, No. 6, pp. 7314-7322).

**Questions:**

1. The authors present accuracy averaged over the last 10 rounds (10% of training). Can they explain the reasoning behind this decision?
2. I'd appreciate access to the code, which is mentioned by the authors, to better understand the implementation details.

---

> ### Author Response · Authors · 2025-11-20
> **Response to Reviewer 1H5i 1/3**
>
> Dear Reviewer 1H5i,
>
> Thank you for your detailed review and thoughtful comments. We believe there may be a few misunderstandings, and we are eager to address your concerns and clarify these potential misunderstandings.
>
> > W1.1: The approach builds upon FedDecorr (replacing the Frobenius norm with log-determinant) and FedETF (incorporating prototypes similar to FedProto/FedNH [3]). Given this foundation, I believe the paper would benefit from either theoretical justification for why this combination is effective or more comprehensive empirical validation.
>
> **A:** Thank you for the insightful comment. We clarify below the novelty of each component and why their combination is effective.
>
> - **LDDecorr.** FedDecorr relies on the Frobenius norm of feature correlation matrix, which treats all singular values uniformly and overlooks the impact of small singular values on dimensional collapse. LDDecorr instead maximizes the log-determinant of the feature correlation matrix, producing **exponential gradients that impose strong penalties on small singular values**. This enables **more effective mitigation of dimensional collapse** than FedDecorr.
>
> - **PBA.** Prior methods such as FedETF and prototype-based alignment (e.g., FedProto, FedNH) address classifier bias or feature inconsistency, but they treat the feature extractor and classifier independently. **PBA uses global prototypes as a shared geometric anchor, jointly aligning features and classifier.** This facilitates the consistency across clients.
>
> - **Synergy of LDDecorr and PBA.** As discussed in the Introduction, the two modules address **distinct yet interdependent challenges** under label skew. LDDecorr prevents dimensional collapse and preserves expressive capacity, which is crucial for generating informative prototypes in PBA. Conversely, PBA imposes structured ETF-like geometry that counteracts potential side effects of strong decorrelation, ensuring that the rank-expansion benefits of LDDecorr translate into improved class separation. Our ablation studies confirm that their combination yields the strongest results.
>
> We have further emphasized these insights in the revised manuscript.
>
> > W1.2: The log-determinant technique for matrix rank minimization is well-established [1], so it would be helpful to clarify the specific novelty here.
>
> **A:** Thank you for the comment. We believe there may be a misunderstanding regarding the role of the log-determinant in our method. While the log-determinant itself is a tool in matrix analysis and rank minimization, **our contribution lies in how it is applied in the context of feature decorrelation**. Specifically, LDDecorr employs the log-determinant of the feature correlation matrix to maximize its rank and impose exponential penalties on small singular values. This directly targets a challenge not addressed by existing FL methods such as FedDecorr. To the best of our knowledge, using a log-determinant objective as a feature decorrelation loss has not been explored in prior FL literature. This point was also noted by Reviewer BH9Y: "The design of the decorrelation loss is quite clever. I don't think I've seen the determinant being used for a decorrelation loss, but this makes sense when we're focusing on the smaller singular values."

---

> ### Author Response · Authors · 2025-11-20
> **Response to Reviewer 1H5i 2/3**
>
> > W1.3: While the methodology section presents various formulas, some appear to go unused in the final approach....
>
> **A:** Thank you for the comment. We would like to clarify the roles of the equations in the methodology section. Some formulas are intermediate steps that support the derivation of our final objectives. For example, Eq. (8) is used to derive the linear penalty of FedDecorr, which is used to highlight the advantage of our exponential penalty. Eq. (10) defines the LogDet divergence between the correlation matrix $\boldsymbol{K}$ and the identity matrix $\boldsymbol{I}$. Building on Eq. (10) and the property $\mathrm{tr}(\boldsymbol{K}) = d$, we obtain the final formulation of LDDecorr. We have revised the manuscript to clarify the roles of these equations and to make these theoretical connections more explicit.
>
> > W1.4: The experimental scope could be broadened to include more diverse heterogeneity scenarios ...
>
> **A:** Thanks for the constructive comment. Since this paper mainly focuses on the label skew, we follow the experimental setting used in prior studies (e.g., FedDecorr [1], FedLC [2]). We agree that exploring broader heterogeneity settings is valuable, such as domain shift or real-world datasets, but these scenarios introduce additional factors beyond the scope of this paper. We view them as important directions for future work.
>
> Regarding model architectures, recent work has shown that pre-trained or BatchNorm-free architectures can mitigate non-IID issues. However, in this paper, our goal is to evaluate whether FedBlade improves robustness without relying on such architectural choices. Therefore, we conduct experiments on MobileNetV2. **We have included additional results on ResNet-18 to broaden the architectural diversity in our evaluation.**
>
> |Method|CIFAR10(0.05)|CIFAR10(0.1)|CIFAR10(0.5)|CIFAR100(0.05)|CIFAR100(0.1)|CIFAR100(0.5)|TinyImageNet(0.05)|TinyImageNet(0.1)|TinyImageNet(0.5)|
> |---|---|---|---|---|---|---|---|---|---|
> |FedAvg|62.09±3.67|73.78±4.11|88.69±0.57|57.08±0.47|60.81±0.44|63.44±0.25|39.70±0.48|43.29±0.25|46.29±0.20|
> |FedProx|62.76±3.97|74.68±3.51|88.65±0.55|56.97±0.33|60.79±0.38|63.19±0.25|39.84±0.40|43.45±0.68|46.38±0.32|
> |FedLC|79.83±0.62|84.68±0.31|89.20±0.17|56.62±0.64|58.77±0.27|61.93±0.20|42.65±0.17|45.66±0.17|47.37±0.21|
> |FedDecorr|67.05±3.33|76.53±3.91|88.74±0.41|54.72±0.38|57.40±0.35|58.14±0.32|40.17±0.46|42.92±0.27|44.83±0.19|
> |FedRCL|63.83±3.02|71.07±1.81|86.84±0.34|52.42±0.31|58.70±0.38|63.49±0.26|37.00±0.33|41.61±0.40|45.53±0.33|
> |FedProto|61.58±3.03|74.41±4.08|89.16±0.47|56.18±0.71|59.69±0.43|64.27±0.25|39.43±0.72|43.53±0.33|47.82±0.24|
> |FedFM|63.79±2.83|75.48±4.73|88.88±0.85|55.43±0.56|61.42±0.68|**67.40±0.24**|34.12±1.25|41.54±0.71|**48.08±0.29**|
> |FedETF|80.41±0.61|85.08±0.38|89.36±0.13|57.61±0.70|60.09±1.21|60.92±1.23|42.79±0.56|45.41±0.18|47.08±0.52|
> |FedBlade|**81.41±0.50**|**85.66±0.13**|**90.25±0.13**|**60.56±0.20**|**63.94±0.27**|65.55±0.13|**43.43±0.28**|**45.84±0.26**|47.94±0.25|
>
> > W2.1: While the focus on dimensional collapse and classifier bias is valuable, ... I'd suggest considering [2] as well, which shares conceptual similarities.
>
> **A:** Thank you for highlighting this related work. FedBR reduces classifier bias by balancing model outputs and learns client-invariant features via a min-max contrastive learning method. Conceptually, it addresses representation inconsistency across clients, similar to prototype-based alignment methods such as FedProto, FedFM, and our PBA module.
>
> However, FedBlade differs in two key aspects. First, FedBlade explicitly address **dimensional collapse**, a phenomenon that FedBR does not target. Second, while FedBR reduces classifier bias via output regularization, FedBlade uses a fixed ETF classifier and **explicitly aligns the feature extractor, projector, and classifier to enforce synergy**. We have included FedBR in the revised manuscript.
>
> > W2.2: Recent work has demonstrated that BatchNorm usage and non-pretrained models ... Happy to provide some starter references on this if needed.
>
> **A:** Thank you for your constructive comment. We have discussed the FL methods focusing on batch normalization in the revised manuscript. Our method is orthogonal to this line of research. FedBlade targets dimensional collapse and classifier bias, which persist even when BatchNorm is replaced or pretrained models are used. Therefore, **it can be integrated with BN-free architectures or pretrained models to further improve robustness**.
>
> **References:**
>
> [1] Shi, Y., Liang, J., Zhang, W., Xue, C., Tan, V. Y., & Bai, S. (2023). Understanding and mitigating dimensional collapse in federated learning. IEEE Transactions on Pattern Analysis and Machine Intelligence, 46(5), 2936-2949.
>
> [2] Zhang, J., Li, Z., Li, B., Xu, J., Wu, S., Ding, S., & Wu, C. (2022, June). Federated learning with label distribution skew via logits calibration. In International Conference on Machine Learning (pp. 26311-26329). PMLR.

---

> ### Author Response · Authors · 2025-11-20
> **Response to Reviewer 1H5i 3/3**
>
> > W3.1: The choice of E=5 local epochs for 100 rounds is interesting....Would a lower E with longer training change the conclusions and result ranking?
>
> **A:** Thank you for your valuable comment. We have conducted an ablation study on the number of local epochs (Appendix E.5). The results show that FedBlade benefits from multiple local updates (performance increases substantially from $E$ = 1 to $E$ = 5). However, the gain from $E$ = 5 to $E$ = 10 is small relative to the additional computation cost. For this reason, we set $E$ = 5 in our main experiments.
>
> |E|CIFAR-10 (α=0.05)|CIFAR-10 (α=0.1)|CIFAR-10 (α=0.5)|CIFAR-100 (α=0.05)|CIFAR-100 (α=0.1)|CIFAR-100 (α=0.5)|Tiny-ImageNet (α=0.05)|Tiny-ImageNet (α=0.1)|Tiny-ImageNet (α=0.5)|
> |---|---|---|---|---|---|---|---|---|---|
> |1|58.33±1.45|64.64±1.51|71.33±0.48|31.90±0.68|31.73±0.53|33.04±0.27|19.03±0.32|18.73±0.56|18.25±0.41|
> |5|75.83±0.70|81.67±0.30|87.90±0.14|54.31±0.19|57.92±0.15|62.07±0.17|39.43±0.17|41.88±0.15|43.63±0.25|
> |10|77.90±0.82|82.63±0.41|88.65±0.22|54.54±0.22|58.30±0.20|61.54±0.28|40.98±0.18|43.22±0.32|44.40±0.31|
>
> Regarding convergence, the results in Appendix E.3 consistently show that FedBlade outperforms all baselines. This confirms that **increasing the number of rounds does not change the ranking among methods**.
>
> > W3.2: Under full participation (Table 6), improvement is marginal, with baselines outperforming or performing within the variance range of the proposed method, despite being simpler. Could you discuss these results?
>
> **A:** Thank you for the helpful comment. Under full client participation, each client has access to more local data and participates in every round, which reduces data heterogeneity and stabilizes prototype generation. **In this setting, the challenges that FedBlade is designed to address, i.e., dimensional collapse and feature inconsistency under label skew, are less pronounced.** Therefore, the gains over simpler baselines naturally become smaller. However, we note that FedBlade remains competitive and does not degrade performance in this setting.
>
> > W3.3: I noticed FedDecorr's is set to 10 here, while their paper used 0.1 under similar settings. Could you clarify this 100x difference to ensure fair comparison?
>
> **A:** Thank you for pointing this out. The value "10" reported in the Appendix D is a typo. In all experiments, we followed the original FedDecorr paper [1] and used $\beta$ = 0.1. The results in the original paper show that when $\beta$ is large, the model performance degrades significantly. We also verified under our own experimental setup that $\beta$ = 10 leads to non-convergence, consistent with the findings reported in [1]. This confirms that such a configuration is not viable and was never used in our experiments. We have corrected this typo in the revised manuscript.
>
> > W3.4: Regarding Figures 5 and 6: the 10% Y-axis increments make the sensitivity appear much lower than it really is, can the authors provide a rendering where the increments are more in line with the performance difference between the proposed method and baselines (e.g. 2% increments)? This would help contextualize the sensitivity better. If test set tuning was used for these hyperparameters, as it appears to be the case, this might disadvantage simpler baselines with fewer hyperparameters. Can the authors comment on this?
>
> **A:** Thank you for the helpful suggestion. We agree that the Y-axis increments in Figures 5 and 6 in the original paper can influence the perceived sensitivity. In the revised manuscript, we have justified the Y-axis increments. The trends remain consistent: FedBlade exhibits moderate sensitivity but maintains stable performance across a range of hyperparameter values.
>
> > Minor corrections: Typo line 108 Relate Work -> Related Work; Typo line 237 definded -> defined
>
> **A:** Thank you for pointing out these typos. We have corrected both ("Related Work" in line 108 and "defined" in line 237) in the revised manuscript.
>
> > Q1: The authors present accuracy averaged over the last 10 rounds (10% of training). Can they explain the reasoning behind this decision?
>
> **A:** We average the accuracy over the last 10 communication rounds to mitigate the stochastic fluctuations that commonly occur in FL training, especially under non-IID settings. This is a standard practice in the literature and provides a more reliable estimate of the converged performance than using a single final-round value, which can vary due to client sampling.
>
> > Q2: I'd appreciate access to the code, which is mentioned by the authors, to better understand the implementation details.
>
> **A:** We appreciate the reviewer's interest. We are finalizing the organization of the code for clarity and reproducibility, and an anonymized link will be provided within the next week for detailed inspection.

---

> ### Author Response · Authors · 2025-11-25
> **Link to Anonymous Repository**
>
> Dear Reviewers,
>
> We provide the link to our implementation (an anonymous repository): https://anonymous.4open.science/r/FedBlade-69FB
>
> We hope this facilitates a thorough and detailed inspection of our methodology and experiments.

---

### Official Review · Reviewer_BbJv · 2025-11-04

**Soundness:** 3
**Presentation:** 3
**Contribution:** 3
**Rating:** 6
**Confidence:** 3

**Summary:**

Data heterogeneity in federated learning, particularly label skew, poses a significant challenge by causing dimensional collapse (where features become low-rank) and classifier bias, ultimately degrading global model performance. To address these issues, this paper proposes FedBlade, a novel framework integrating two key components: LDDecorr and PBA. LDDecorr is a feature decorrelation method that maximizes the log-determinant of the feature correlation matrix; this yields exponential gradients that, unlike previous linear methods, apply an "infinite penalty" to small singular values, effectively and rapidly mitigating dimensional collapse. PBA (Prototype-guided Bidirectional Alignment) enhances the synergy between the model's feature extractor, projector, and a fixed ETF classifier by using global prototypes as a common reference, ensuring these modules are aligned and consistent across clients. Extensive experiments show that FedBlade outperforms existing baselines in accuracy and achieves substantially faster convergence.

**Strengths:**

1. The paper proposes LDDecorr, a novel feature decorrelation method. Unlike previous approaches like FedDecorr that use linear gradients, LDDecorr uses a log-determinant formulation to produce exponential gradients ($\nabla_{\lambda_{i}} = -1/\lambda_{i}$). This is a significant strength because it imposes an "infinite penalty" on small singular values, preventing them from collapsing to zero much more effectively and accelerating the mitigation of dimensional collapse.

2.  The paper identifies a key weakness in prior work (like FedETF) that uses fixed classifiers, which lacks synergy between the feature extractor, projector, and classifier. Its second component, PBA (Prototype-guided Bidirectional Alignment), directly solves this. It uses global prototypes as a common "bridge" to simultaneously align the feature extractor with the prototypes and the projector with the fixed classifier, ensuring all model parts work together coherently.

3. FedBlade consistently outperforms a wide range of relevant baselines (including FedAvg, FedDecorr, and FedETF) in final accuracy, especially on complex datasets like CIFAR-100 and Tiny-ImageNet. Furthermore, it shows FedBlade achieves substantially faster convergence.

**Weaknesses:**

1. Feature decorrelation method (LDDecorr) is sensitive to the decorrelation strength,.

2.  The proposed PBA method requires an extra communication step in each round. Clients must compute and send their local prototypes to the server, and the server must aggregate the global prototypes and send them back to the clients. This adds to communication costs, a key bottleneck in federated learning.

3. The FedBlade framework adds computational overhead on both the client and server. Clients must perform extra calculations for the new loss terms: $\mathcal{L}_{LDDecorr}$ which involves a determinant calculation and the two PBA losses. The server also has the new task of aggregating all client prototypes.

**Questions:**

1. What are the limitations of the proposed approach?

2.  How do the computational costs of LDDecorr and the communication costs of PBA scale, especially with high-dimensional feature spaces and a large number of classes? At what point do these added costs make FedBlade less practical than faster, simpler methods?

---

> ### Author Response · Authors · 2025-11-20
> **Response to Reviewer BbJv**
>
> Dear Reviewer BbJv,
>
> Thank you for valuable comments and recognition of our paper. Below we address specific questions:
>
> > W1: Feature decorrelation method (LDDecorr) is sensitive to the decorrelation strength.
>
> A: Thanking you for pointing this out. LDDecorr introduces an exponential penalty on singular values, which accelerates the suppression of small singular values and effectively mitigates dimensional collapse. This design may make the method more sensitive to the regularization strength. Sensitivity analysis in Appendix D shows that LDDecorr performs robustly within a reasonable range of strengths, though overly large values can lead to over-penalization. Exploring alternative formulations that preserve fast collapse mitigation while reducing hyperparameter sensitivity is an interesting direction for future work.
>
> > W2: The proposed PBA method requires an extra communication step in each round.... This adds to communication costs, a key bottleneck in federated learning.
>
> A: Thank you for raising this concern. We would like to clarify that **the extra communication from PBA is negligible compared to model transmission**. Each client sends one 1280-dimensional float32 vector per class (≈5 KB per class), whereas MobileNetV2 contains ≈3.4M parameters (≈13.6 MB). Thus, even for datasets with many classes, prototype exchange requires only a tiny fraction of the per-round bandwidth. We have added this quantitative analysis to the revised manuscript.
>
> |Model|CIFAR-10|CIFAR-100|Tiny-ImageNet|
> |---|---|---|---|
> |MobileNetV2|50KB|500KB|1000KB|
> |ResNet-18|20KB|200KB|400KB|
>
> > W3: The FedBlade framework adds computational overhead on both the client and server..... The server also has the new task of aggregating all client prototypes.
>
> A: Thanking you for pointing this out. Computing the LDDecorr term involves the determinant of a $d \times d$ SPD matrix, which can be obtained via Cholesky decomposition in $\frac{1}{3} d^{3}$ FLOPs. Prototype aggregation on the server is simple averaging and incurs negligible overhead. To quantify the practical cost, we report the per-iteration computation cost (ms) of each loss term:
>
> |Dataset|Model|$\mathcal{L}_{sup}$|$\mathcal{L}_{FA}$|$\mathcal{L}_{PA}$|$\mathcal{L}_{LDDecorr}$|
> |---|---|---|---|---|---|
> |CIFAR-10|MobileNetV2|0.0886|0.1824|0.1294|1.1522|
> |CIFAR-10|ResNet-18|0.0882|0.1551|0.1255|0.5297|
> |CIFAR-100|MobileNetV2|0.0933|0.2566|0.1373|1.1600|
> |CIFAR-100|ResNet-18|0.0912|0.1606|0.1299|0.5592|
> |TinyImageNet|MobileNetV2|0.0884|0.5700|0.1322|1.1460|
> |TinyImageNet|ResNet-18|0.0899|0.1947|0.1282|0.5245|
>
> These results show that LDDecorr adds roughly 0.5–1.2 ms per iteration. Thus, **the additional computations introduced by FedBlade are modest and do not create a practical bottleneck**. We have included this analysis in the revised manuscript.
>
> > Q1: What are the limitations of the proposed approach?
>
> A: We acknowledge that our approach has several limitations. First, LDDecorr is sensitive to its regularization strength. Although the value used in our experiments ($\beta$ = 0.005) performs robustly across architectures and datasets, overly large strengths can dominate the learning objective and hinder representation learning. Developing decorrelation methods that retain strong collapse mitigation while being less sensitive is an important direction for future work. Second, PBA requires transmitting class prototypes between clients and the server. The communication cost is negligible relative to model transmission, but designing alignment mechanisms that avoid additional prototype exchange would further improve efficiency. We plan to explore these directions in future work.
>
> > Q2: How do the computational costs of LDDecorr and the communication costs of PBA scale, ... At what point do these added costs make FedBlade less practical than faster, simpler methods?
>
> A: Thank you for the thoughtful question on scalability. For LDDecorr, the dominant cost is computing the determinant of a $d \times d$ representation correlation matrix, which requires $\frac{1}{3} d^{3}$ FLOPs via Cholesky decomposition. Under our experiments (512 or 1280 feature dimensions), this adds only 0.5–1.2 ms per iteration (see W3). The cost grows cubically with $d$, so LDDecorr may become less practical only when using extremely high-dimensional feature spaces, which are uncommon in modern architectures used in FL.
>
> For PBA, the communication cost scales linearly with the number of classes $C$ and feature dimension $d$, i.e., $4Cd$ bytes. Even for datasets like Tiny-ImageNet (200 classes), the total per-round overhead is only ~1 MB, which is negligible compared to the transmission for model parameters.
>
> In summary, **within the model and dataset scales typical in federated learning, the costs of LDDecorr and PBA remain small and do not hinder practicality.** Moreover, FedBlade converges in fewer rounds, **further reducing overall computation and communication** relative to standard FL baselines.

---

### Author Response · Authors · 2025-11-24
**General Response to All Reviewers**

We sincerely thank all reviewers for their thoughtful and constructive feedback. We are encouraged that reviewers recognized the motivation of this work, the strength of methodology, and the clarity of presentation. In particular, we appreciate comments such as:

- "The paper benefits from a clear motivation, and the target problem is an important one." (Reviewer 1H5i)
- "The motivation and intuition of this work are clear and sound, and theoretical analysis connecting the log-determinant term to spectrum isotropy and effective rank is insightful." (Reviewer CBFg)
- "The authors do well in emphasizing the need to focus on the small singular values." (Reviewer BH9Y)
- "LDDecorr is a significant strength." (Reviewer BbJv)
- "The design of the decorrelation loss is quite clever." (Reviewer BH9Y)
- "Using global prototypes as bridges between modules is a natural and elegant idea that improves interpretability." (Reviewer CBFg)

The reviewers’ constructive feedback has helped us strengthen the paper. We addressed the comments in the individual responses and updated the paper accordingly, with key changes highlighted in blue. In summary, the main revisions are outlined below:

**1. Synergy of LDDecorr and PBA.**
- We expanded the discussion of how the two modules address complementary aspects of label skew and why their interaction is essential (see **Introduction** and **Section 5.3**).

**2. Analysis of computation and communication cost.**
- We added detailed computation overhead for each loss component on multiple model architectures and datasets (see **Appendix E.6**).
- We added the communication cost of PBA across different model architectures and datasets (see **Appendix E.7**).

**3. Additional experimental results.**
- We added results on ResNet-18 to broaden architectural diversity (see **Appendix E.2**).
- We added the quantitative analysis of neural collapse (see **Section 5.3**).
- We added the effective rank of all methods (see **Section 5.3** and **Appendix E.4**).
- We added the sensitivity analysis of temperature (see **Appendix D**).
- We added the ablation on the number of local epochs (see **Appendix E.5**).
- We expanded the discussion on the experimental results (see **Section 5.3** and **Appendix E.1**.)

We sincerely thank the reviewers for their insightful comments, which helped improve the clarity and completeness of this paper. We hope that these revisions and responses effectively address all concerns.

---

### Author Response · Authors · 2025-12-01
**Summary for the Area Chair**

We sincerely thank the Area Chair for taking the time to review our paper and rebuttal.
Below, we summarize how our rebuttal directly addresses the reviewers’ concerns.
For an overview of contributions recognized by all reviewers and the revisions made during the discussion phase, please refer to the comment **[General Response to All Reviewers]**.

---

**Reviewer BbJv:**

- **W1: Sensitivity of LDDecorr.**
We clarified why LDDecorr is inherently more sensitive due to its exponential penalty on small singular values, and we demonstrated that it remains robust within a reasonable range.

- **W2 & Q2: Communication cost of PBA.**
We quantified prototype exchange costs across architectures and datasets and showed that *the extra communication is negligible* compared to model transmission.

- **W3 & Q2: Computation cost of LDDecorr and PBA.**
We added detailed computation overhead for every loss component. These results show that *the additional overhead is small and does not introduce a practical bottleneck*.

- **Q1: Limitations of FedBlade.**
We explicitly discussed limitations, including sensitivity of LDDecorr and the requirement for prototype exchange.


**Reviewer 1H5i:**

- **W1.1: Justification for combining LDDecorr and PBA.**
We highlighted the novelty of each module and clarified their synergy: LDDecorr expands feature rank while PBA stabilizes the geometric structure.

- **W1.2: Novelty of LDDecorr.**
The reviewer provided a work published in **the Proceedings of the 2003 American Control Conference**, which uses log-determinant for matrix rank minimization. We clarified that while the log-determinant is a known mathematical tool, the novelty lies in using it as a decorrelation loss for feature representations in FL, which has not appeared in prior FL literature.

- **W1.3, W3.3, W3.4 & Minor corrections.**
We refined and clarified the manuscript accordingly.

- **W1.4 & W3.1: Additional experiments.**
We added new experiments including broader architectures and an ablation on local epochs.

- **W2.1 & W2.2: Related work.**
We incorporated the suggested literature and clarified conceptual distinctions.

- **W3.2: Interpretation of full-participation results.**
We explained why improvements are smaller under low heterogeneity.

- **Q1: Averaging over last 10 rounds.**
We justified this as a standard method to reduce stochasticity in FL training.

- **Q2: Access to code.** We provided an anonymous repository: https://anonymous.4open.science/r/FedBlade-69FB.

**Reviewer CBFg:**
- **W1: Methodology clarity.**
We clarify the contribution of LDDecorr and PBA, and explain that LDDecorr and PBA provide *complementary capabilities* rather than incremental additions.

- **W2, W4 & Q1: Interpretation of experimental results.**
We expanded analysis in the manuscript and explained behaviors across datasets, heterogeneity levels, and training phases.

- **W3 & Q2: Synergy of LDDecorr and PBA.**
We provided deeper analysis showing that *the two modules solve distinct but interdependent challenges under label skew*.

- **W5: Testing on additional architectures.**
We added results on ResNet-18, which show consistent gains.

- **W6: Cost metrics.**
We provided both computation and communication cost analyses and clarified that FedBlade reduces total training time by converging in fewer rounds.

- **Q3-Q6: Additional experimental results.**
We added these results as suggested, including convergence curves, sensitivity analyses, communication overhead, and neural-collapse metrics.

**Reviewer BH9Y:**
- **W1 & Q1: Effective rank comparisons.**
We added effective rank results for all methods and clarified their relationship to accuracy. The new analysis highlights that *effective rank and accuracy are not strictly monotonic*, and demonstrates how LDDecorr and PBA jointly improve both representation diversity and geometric structure.

- **W2 & W3: Related work.**
We incorporated the missing references and clarified differences from FedUV and FedBABU.

- **W4: Naming of "bidirectional alignment".**
We adopted "bilateral alignment", which more accurately describes its function.

- **Q2: Methodology clarity.**
We further differentiated our approach through analysis of *spectral structure and the role of prototypes*.

- **Reviewer acknowledgement and remaining concern.**
Reviewer BH9Y explicitly noted: "I appreciate the new analysis… it is intuitive that having a higher rank is not always good." The only remaining concern was that some explanations were not sufficiently visible in the main text.
To fully address this, we revised the main text to clearly state why effective rank and accuracy are not monotonic and how this relates to our method’s design.

---

For additional details, we kindly invite the Area Chair to refer to the full rebuttal, where all explanations, analyses, and newly added results are provided comprehensively.

---

### Meta-Review · Area_Chair_pNQc · 2026-01-04

**Summary:**

This paper proposes a federated learning framework with bilateral alignment and feature decorrelation to address the challenges of dimensional collapse and classifier bias induced by data heterogeneity. Although the authors provided comprehensive responses and revisions during the rebuttal, the core criticisms raised by the reviewers point to fundamental weaknesses in the paper's novelty, methodological necessity, and experimental depth.

**Reviewer Concerns:**

The reviewer did not participate in the discussion during the discussion stage. However, the reviewers have the following commonalities in their initial comments:

**Lack of Novelty:** Multiple reviewers (particularly 1H5i and CBFg) explicitly pointed out that the proposed method (FedBlade) essentially combines existing works—specifically, an improved version of FedDecorr (LDDecorr) and enhanced components from FedETF/FedProto (PBA). As a result, its core innovation was perceived as limited.

**Limitations in Experimental Results:** Reviewers CBFg and 1H5i noted that under milder heterogeneity settings (such as full client participation), FedBlade's performance improvements were marginal, with some baselines performing on par. This undermines the method's effectiveness and necessity in broader scenarios. Additionally, issues were identified in the ambiguity of ablation studies and the justification regarding effective rank.

**Narrow Experimental Scope:** Reviewers CBFg and 1H5i considered the experimental scope limited, as it primarily relied on MobileNetV2 and a small set of image datasets, lacking validation across more architectures (e.g., BN-free networks or pre-trained models) and non-vision domains. Although the authors supplemented results with ResNet-18, this did not fully alleviate concerns regarding the generalizability of the method.

Regarding the issue of novelty in the paper, I agree with the reviewer's suggestion that the innovation of this paper is incremental. In the experimental section, the calculation and communication cost analysis provided by the author, as well as the addition of ablation experiments, have to some extent dispelled the concerns of the reviewers. However, regarding the comparison of effective rank with other methods, I believe there are still issues: effective rank and accuracy are not strictly monotonic. This indicates a logical loophole in the core argument of the paper (improving performance by increasing effective rank), which weakens the theoretical foundation of the method.

**Reviewer Scores:**

Despite the authors' detailed responses, the core points of contention remain unresolved. Reviewers are unlikely to significantly adjust their scores.
**Reviewer BbJv** raised concerns about computational/communication overhead and methodological sensitivity; while the authors provided data, this did not fundamentally alleviate their concerns.
**Reviewer 1H5i**, with a score of 2 and a self-assessed confidence level of 5, directly questioned the paper's fundamental novelty, arguing it was merely a simple combination of existing work.
**Reviewer CBFg** raised numerous experimental questions, and the authors failed to address these core weaknesses. For instance, the fact that "limited improvement under full participation" remains unchanged, and the authors' explanations implicitly acknowledge the method's limitations.
**Reviewer BH9Y**, who clearly commented, will maintain his score, precisely because the authors' responses did not meet his expectations.

---

### Decision · Program_Chairs · 2026-01-26

Reject